# Computation and data driven discovery of topological phononic materials

Jiangxu Li[1,2], Jiaxi Liu[1,2], Stanley A. Baronett [3], Mingfeng Liu[1,2], Lei Wang [1,2], Ronghan Li[1], Yun Chen [1,2], Dianzhong Li[1,2], Qiang Zhu [3✉] & Xing-Qiu Chen [1,2✉]

The discovery of topological quantum states marks a new chapter in both condensed matter physics and materials sciences. By analogy to spin electronic system, topological concepts have been extended into phonons, boosting the birth of topological phononics (TPs). Here, we present a high-throughput screening and data-driven approach to compute and evaluate TPs among over 10,000 real materials. We have discovered 5014 TP materials and grouped them into two main classes of Weyl and nodal-line (ring) TPs. We have clarified the physical mechanism for the occurrence of single Weyl, high degenerate Weyl, individual nodal-line (ring), nodal-link, nodal-chain, and nodal-net TPs in various materials and their mutual correlations. Among the phononic systems, we have predicted the hourglass nodal net TPs in $TeO_3$, as well as the clean and single type-I Weyl TPs between the acoustic and optical branches in half-Heusler LiCaAs. In addition, we found that different types of TPs can coexist in many materials (such as ScZn). Their potential applications and experimental detections have been discussed. This work substantially increases the amount of TP materials, which enables an in-depth investigation of their structure-property relations and opens new avenues for future device design related to TPs.

[1] Shenyang National Laboratory for Materials Science, Institute of Metal Research, Chinese Academy of Sciences, Shenyang, China. [2] School of Materials Science and Engineering, University of Science and Technology of China, Shenyang, China. [3] Department of Physics and Astronomy, University of Nevada, Las Vegas, NV, USA. ✉email: qiang.zhu@unlv.edu; xingqiu.chen@imr.ac.cn

Over the past decade, topological concepts have made far-reaching impacts on the theory of electronic band structures in condensed matter physics and materials sciences[1–3]. Thousands of topological electronic materials[4–8] were theoretically proposed[9–11] and some of them were experimentally verified, such as, topological insulators[4–8], Dirac/Weyl semimetals[12–16], and nodal-line semimetals[17–21]. As the counterpart of electrons, phonons[22] are energy quanta of lattice vibrations. They make crucial contributions to many physical properties, such as, thermal conductivity, superconductivity, and thermoelectricity, as well as specific heat. Similar to topological electronic nature, the crucial theorems and concepts of topology can be introduced to the field of phonons, called topological phononics (TPs)[23–43]. In particular, TPs in solid materials are also correlated to some specified atomic lattice vibrations generally within a scale of THz frequency, thereby providing a rich platform for the investigation of various quasiparticles related with Bosons.

TPs have been theoretically or experimentally investigated in solid-state materials. Several theoretical models, including monolayer hexagonal lattices[35,36], Kekulé lattice[37,38], and one-dimensional (1D) chains[39], were discussed. More recently, a number of real materials were predicted to host the Weyl TPs[23–29,33], nodal-line TPs [30–34], and nodal-ring phonons[32]. Single Weyl TPs were predicted in noncentrosymmetric WC-type materials[23,25], exhibiting twofold degenerate Weyl points with the ±1 topological charges. In FeSi-type materials, double Weyl TPs were predicted and then experimentally confirmed[24,40]. In $SiO_2$, the coexisted single and double Weyl TPs were suggested[29]. In addition to occupying the discrete sites in the reciprocal space as Weyl points, these band crossing points can also continuously form nodal-lines (e.g., in $MgB_2$ (ref. [31]) and $Rb_2Sn_2O_3$ (ref. [33])) or nodal-rings TPs (e.g., in graphene[32], bcc $C_8$ (ref. [30]), and $MoB_2$[34]). TPs exhibit the typical features of bulk-surface (edge) correspondence, which are rooted in different geometry phases of Hamiltonian. The existence of multiple critical physical phenomena, such as phononic valley Hall effect[36], phononic quantum anomalous Hall-like effect, and phononic quantum spin Hall-like effect controlled by multiple-valued degrees of freedom[37], are beneficial to TPs' applications. Because the topologically protected states are immune to backscattering[1–3], TPs would be very promising for applications in the abnormal heat transport[44,45], and phonon waveguides[38], and so on. In addition to the atomic crystals, topological phonons have also been extensively studied in mechanical metamaterials[46–50], acoustic systems[51–53], and Maxwell frames[54–57].

Unlike the topological electronic materials in which one only needs to focus on energy states near the Fermi level, phonons exhibit several distinct properties. First, there are no limits of Pauli exclusion principle. Second, each phonon mode, following Bose–Einstein statistics, can become practically active, due to thermal excitation. Third, phonons are charge neutral and spinless Bosons, which can not be directly influenced by the electric and magnetic fields. Hence, a full frequency analysis for all phononic branches is needed for the study of TPs. To date, a large-scale identification of TP materials remains challenging, because it is far more expensive to compute the phonon band dispersions than to calculate the electronic band structures. Hence, it is certainly more difficult to seek feasible TP materials in a high-throughput (HT) computational manner, as compared with the recent works in topological electronic materials[9–11].

Herein, we present an efficient and fully automated workflow that can screen the TP crossings in a large number of solid materials. Our results reveal that TPs extensively exist in phonon spectra of many known materials, which can be classified into two main types of Weyl and nodal-line (ring) TPs. We have

elucidated the physical mechanism for the occurrence of Weyl and nodal-line (ring) TPs. Weyl TPs can be found as (1) single TPs in the crystals without the inversion symmetry, and (2) high degenerate TPs in the noncentrosymmetric crystals with the presence of screw rotations. Nodal-line (ring) TPs can be found as individual nodal-line (ring), nodal-link, nodal-chain, and nodal-net TPs in the crystals with $PT$ symmetry, upon the manipulation of the nonsymmorphic symmetry elements (e.g., screw rotation and glide reflection). Among the phononic systems, we have predicted the hourglass nodal-net (HNN) TPs in $TeO_3$, as well as the clean and single type-I Weyl TPs between the acoustic and optical branches in half-Hesuler LiCaAs. We also found that the extensive coexistence of different types of TPs in materials, such as, the coexisted threefold and fourfold degenerate Weyl TPs in BeAu and the coexisted nodal-line and nodal-ring TPs in ScZn.

## Results

**High-throughput screening strategy of topological phonons.** As shown in several prototypical materials[23–42], identifying TPs requires several stages of manual selections and subjective human decisions. To enable the TPs discovery in an automatic manner, we present a HT screening and data-driven approach to discover and categorize TPs, as described in Fig. 1, including the following four steps.

(1) *Phonon data collection.* To obtain phonon spectra for a large volume of known materials, we first collected 10,000 materials' force constant data from public phonon database[58,59]. The data set was further augmented by our in-house computations for over 2000 materials belonging to 58 common structural prototypes. It is well known that the calculated force constants are numerically sensitive to the choices of several parameters, e.g., the supercell size, $K$-point mesh, and energy cutoff. To guarantee that the predictions are reliable, we filtered out the materials with notable imaginary frequencies (<−0.5 THz) in the whole phononic momentum space.

(2) *Nodal straight lines identification.* We computed their band dispersions on the automatically generated high-symmetry band paths[60]. If there exist degenerate phononic bands along high-symmetry paths, we would compute the Berry phase $\gamma_n = \oint_c dl \cdot \mathcal{A}_n$, by an integral of Berry connection ($\mathcal{A}_n(q) = i < \mu_{n,q}|\nabla_q|\mu_{n,q}>$) over a closed $q$ path[61], for 20 consecutive points on each of these bands. The bands possessing continuous points with Berry phase values, ±π, would be marked as the topologically nontrivial nodal straight lines.

(3) *Crossing points screening.* For the rest band paths in the phonon spectra, we systematically scanned 50 points on each band path. In the entire frequency range, we considered the points possessing two adjacent eigenfrequencies < 0.5 THz. For each point, we performed a minimization based on the conjugate gradient algorithm to obtain the local minimum of the frequency difference ($\Delta_{freq}$). The points with $\Delta_{freq}$ < 0.2 THz were checked if they possess with Berry phase values of ±π. After optimization, the identified crossing points may go anywhere in the entire reciprocal space. Therefore, we also checked if the points are at or off the high-symmetry paths.

(4) *Crossing points assignment.* The identified phononic crossing points were then divided into two groups based on the presence of both inversion symmetry ($P$) and TRS ($T$) for each material. In a three-dimensional (3D) system with $PT$ symmetries, the Berry curvatures of nondegenerate

## Workflow for high-throughput screening of topological phonons

**Fig. 1 The schematic flowchart of high-throughput computational screening on topological phonons.** This workflow is capable of identifying the features of TPs, elucidating details of topology and constructing TPs database by computing and collecting phonons of a variety of solid materials in an automatic manner.

phononic bands are forced to be zero and the Weyl TPs would not occur in such system. Once the phononic bands at a degenerate point have opposite nonzero Berry phases, such topological nontrivial degenerate points have to occur continuously by forming nodal-line (ring) TPs, due to the continuity of phonon wave function in the 3D momentum space. As a result, when the $PT$ symmetry is present, we just need to seek nodal-lines (rings) off high-symmetry line. In noncentrosymmetic materials, the phonon dispersions possibly form single Weyl or high degenerate Weyl TPs, in addition to nodal-lines (rings) TPs. In order to clarify these three types of TPs, we introduced another formula of Chern number, which can be derived by integrating Berry curvatures of a closed surface[61] according to $n = \frac{1}{2\pi} \int_S d\mathbf{S} \cdot \Omega(\mathbf{q})$. Here, $\mathbf{S}$ is a closed surface which wraps the target crossing point, and the $\Omega(\mathbf{q})$ is the Berry curvature at the phonon momentum $\mathbf{q}$ on the selected closed surface. For the isolated crossing points at the high-symmetry band paths, we marked the points with nonzero integer Chern numbers (e.g., ±1, ±2) as single or high degenerate Weyl points. Otherwise, they would be labeled as nodal-ring points, given the fact that the nodal-line points were already extracted in step (2). Of course, it needs to be emphasized that many materials may yield multiple crossing bands along off high-symmetry paths, which can be separate Weyl TPs or nodal-line (ring) TPs (Supplementary Table 1). In principles, this approach can be easily extended to investigate these crossing points.

In order to experimentally observe phononic surface states, a TP material is expected to possess distinct Weyl or nodal-line (ring) TPs and possible clean nontrivial surface TPs. For this purpose, we mathematically define the clean TPs for the nontrivial crossing points satisfying two conditions in bulk phonon spectra, (i) the crossing points have to be located at local minima with negligible or zero phononic density of states (DOS; <0.01 states/atom/THz) and (ii) the dispersion at the local minima is sufficiently large ($\partial E/\partial q > 3.0$ THz · Å). On basis of these two criteria, we have filtered 322 clean TP materials

(Supplementary Table 5). For instance, single type-I Weyl TPs in LiCaAs are clean, because it does not overlap with the other bulk phonon branches and has a zero phonon density at the frequency of 4.590 THz (Fig. 2i, j).

**Topological phonons of materials**. In total, our approach revealed that 5014 materials exhibit TPs states (Supplementary Table 1). Among them, we have identified two main categories of nodal-line (ring) and Weyl TP materials. Among nodal-line (ring) TPs (Supplementary Table 2), there are several possible subclasses, including nodal-link, nodal-chain, and nodal-net TPs according to different symmetries. Among Weyl TPs, there are two main subclasses of single Weyl TPs (Supplementary Table 3) and high degenerate Weyl TPs (Supplementary Table 4) according to the degree of phononic band degeneracy. In particular, we note that, different from electronic system, it is impossible to have the intrinsic TP insulators without any tunable external field. This is mainly because the phonon spectrum always preserves the time-reversal symmetry (TRS). The TRS for phonon spectrum was suggested to be broken only in artificial lattices, such as in ionic lattices using Lorentz force[35] and magnetic lattices using spin–lattice interactions[62]. Therefore, we did not attempt to identify the intrinsic topological phonon insulators in our current work.

To understand the occurrence of TPs in various materials and their correlations, we chose to investigate four representative cases, including (1) half-Heusler LiCaAs alloy for single Weyl TPs, (2) superconducting BeAu for high degenerate Weyl TPs, (3) ScZn with the $PT$ symmetries for nodal-line (ring) TPs, and (4) TeO$_3$ with both $PT$ and nonsymmorphic symmetries for HNN TPs. To conveniently analyze their TP states, we use a general $\mathbf{k} \cdot \mathbf{p}$ Hamiltonian as follows,

$$H(\mathbf{q}) = \sum_{i=0}^{3} d_i(\mathbf{q})\sigma_i \qquad (1)$$

in which $\sigma_0$ is a $2 \times 2$ identity matrix and $\sigma_{i=x,y,z}$ denote the Pauli matrices, respectively, $d_i(\mathbf{q})$ are real functions, and $\mathbf{q} = (q_x, q_y, q_z)$

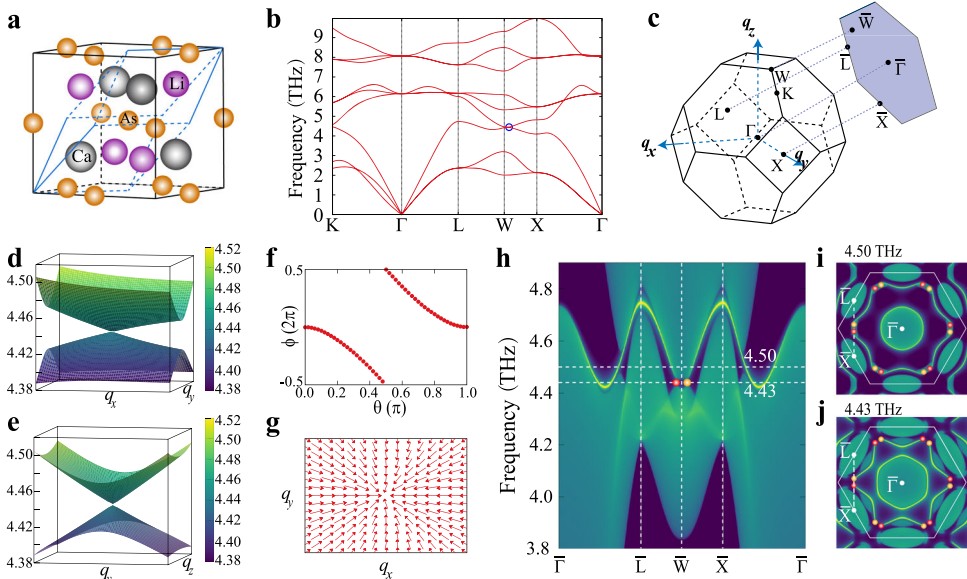

**Fig. 2 Phonon dispersion and single Weyl TPs of LiCaAs. a** The unit cell and primitive cell of LiCaAs (space group $F\bar{4}3m$ 216). **b** The phonon dispersion along the high-symmetry lines. The blue circles denote the phononic crossing points (the single Weyl TPs). **c** The bulk BZ and the (111) surface BZ. **d, e** The 3D phonon dispersions centered at the Weyl point on the $q_{xy}$ and $q_{yz}$ planes, respectively. **f, g** The Wannier center evolution and Berry curvatures distributions around this Weyl phonon. **h** The surface phononic states along the high-symmetry lines. The red and yellow circles represent the projected Weyl TPs with positive and negative topological charges of 1 and $-1$, respectively. **i, j** the Weyl phonon induced nontrivial surface phononic arc states at 4.50 and 4.43 THz, respectively.

are wave vectors of phonons. The energy spectrum is

$$\omega(\mathbf{q}) = d_0 \pm \sqrt{d_1^2 + d_2^2 + d_3^2} \qquad (2)$$

Applying different constraints of crystal and $PT$ symmetries to this Hamiltonian, we can describe various geometries of the phonon crossings.

**Single Weyl TPs in noncentrosymmetric half-Heusler alloys.** Half-Heusler compounds $A^{I}B^{II}X^{V}$ (where $A \in \{Li, Na, K, Rb, Cs\}$; $B \in \{Mg, Ca, Zn\}$, and $X \in \{P, As\}$) with non-centrosymmetric space group $F\bar{4}3m$ (no. 216) have been widely studied [63,64]. Since they share similar phonon dispersions, here we focus on the case of LiCaAs. As shown in Fig. 2a, the As atoms are in the $4a$ Wyckoff sites, while both Li and Ca are in the $8c$ sites. At 4.590 THz, the highest longitudinal acoustic (LA) and the lowest transverse optical (TO) branches have a band cross at the $(0.435, 0.0, 1.0)\frac{2\pi}{a}$ point along the $W$–$X$ high-symmetry path Fig. 2b. Interestingly, the phononic spectra around this crossing point exhibit a Weyl cone-like shape in both $q_{xy}$ and $q_{yz}$ planes, as shown in Fig. 2d, e. To further confirm its topological nature, the Wannier center evolution has been derived for the third phononic band (Fig. 2f). It gives a topological charge of $-1$ and this crossing point acts like the sink of Berry curvatures in Fig. 2g. These results indicate that the crossing point along the $X$–$W$ path is an ideal type-I Weyl TP with the topological charge of $\pm 1$.

To elucidate the underlaying physics for the type-I Weyl TPs in LiCaAs, we constructed the Hamiltonian from Eq. (1) according to the fact that LiCaAs possesses both the TRS $T$ and the twofold rotational symmetry $C_2$. As shown in Fig. 2d, e, the type-I WPs do not have the tilt term (expressed as $d_0\sigma_0$ in Eq. (1)). Combining $T$ and $C_2^z$ symmetries, the operator can be represented by $\sigma_z\kappa$ (where $\kappa$ is the complex conjugate operator). The Hamiltonian

$$H(-C_2^z\mathbf{q}) = C_2^z TH(\mathbf{q})T^{-1}C_2^{z-1}, \qquad (3)$$

should be subject to the following constraints.

$$\begin{aligned} d_1(q_x, q_y, -q_z) &= -d_1(q_x, q_y, q_z), \\ d_{2,3}(q_x, q_y, -q_z) &= d_{2,3}(q_x, q_y, q_z), \end{aligned} \qquad (4)$$

where they require $q_z = \frac{2\pi n}{a}$. When $q_z = \frac{2\pi}{a}$ (see Fig. 2c), the crossing points on the square plane at the BZ's boundary satisfy Eq. (4) and $d_1$ has to be 0. We further consider the other two rotational symmetries, $C_2^x$ and $C_2^y$, at the condition of $q_z = \frac{2\pi}{a}$. It gives

$$d_{2,3}(q_x, -q_y) = -d_{2,3}(q_x, q_y) \qquad (5)$$

or

$$d_{2,3}(-q_x, q_y) = -d_{2,3}(q_x, q_y) \qquad (6)$$

This condition implies that the crossing points on the square plane ($q_z = \frac{2\pi}{a}$) are constrained to the high-symmetry lines along either $q_x$ or $q_y$ direction. As a result, the type-I Weyl TPs on the $W$–$X$ line of the BZ boundary are protected by both $C_2$ and $T$ symmetries, which can produce well-separated WPs and give rise to the extremely long open arcs on the surface. The topologically protected surface states of (111) surface along high-symmetry line have been derived in Fig. 2h and the Weyl points marked as the red and yellow spheres are projected on the $\bar{L}$–$\bar{X}$ line. The open surface arcs have also been clearly observed at 4.43 and 4.50 THz (Fig. 2i, j), respectively. Each pair of projected Weyl points with opposite charges is clearly connected by the open surface arc, which is long enough and can provide a robust one-way phonon propagation channel on the surface without backscattering from defects. To our best knowledge, the Weyl TPs in LiCaAs exhibit two unique features different from the known Weyl TPs. First, this type-I Weyl point is the only phononic band crossing between the LA and TO branches. Second, it is the only known clean Weyl TP, which doesn't overlap with any other phonon branches. The calculations further reveal that the family of half-Heusler $A^{I}B^{II}X^{V}$ compounds host similar Weyl TPs and their nontrivial long arc surface states. These materials are suitable for

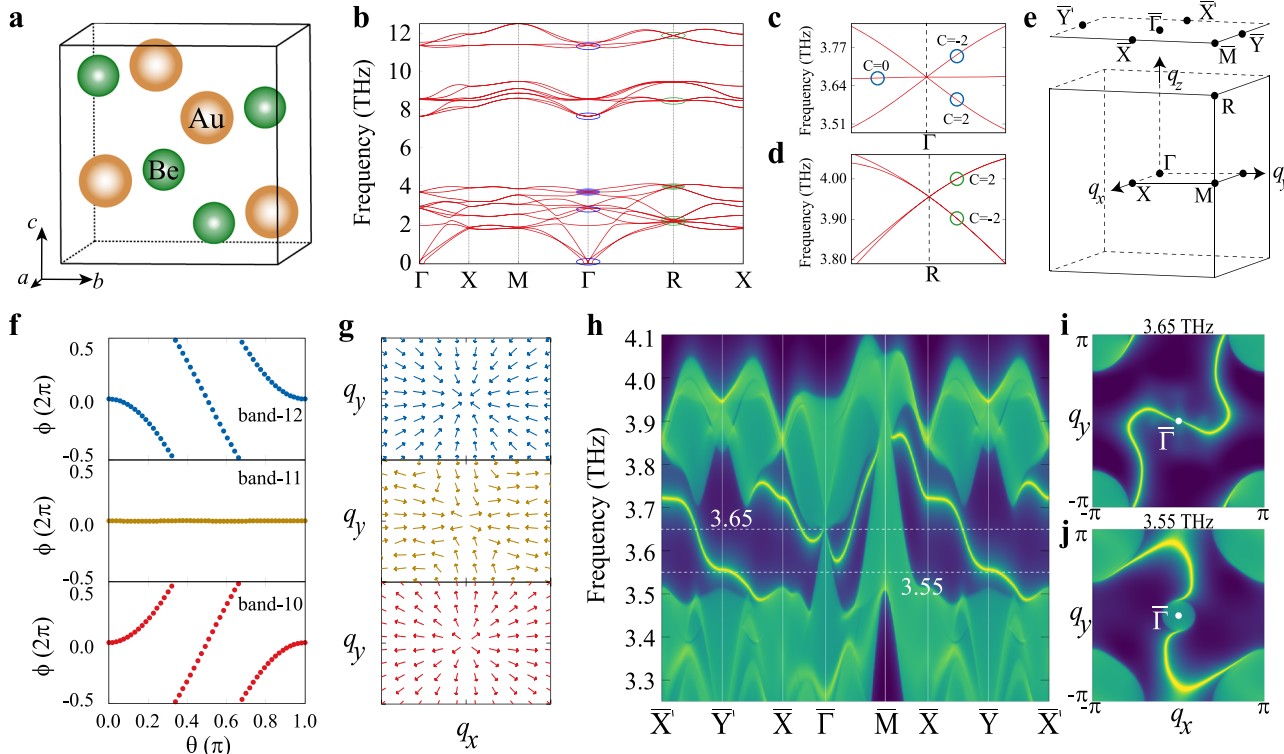

**Fig. 3 Phonon band structures and surface states for topological high degenerate Weyl TPs in BeAu. a** The crystal structure of BeAu (space group $P2_13$ 198). **b** The phonon dispersion along the high-symmetry lines. The blue circles are the threefold degenerate Weyl TPs at $\Gamma$ and the green circles are the fourfold degenerate Weyl TPs at $R$. **c** Threefold degenerate Weyl point of 3.669 THz at $\Gamma$. **d** Fourfold degenerate Weyl TPs of 3.956 THz at $R$. **e** Bulk and surface BZ of BeAu. **f** The Wannier center evolution for three branches 10, 11, and 12 centered at the $\Gamma$. **g** The Berry curvature distributions of three branches 10, 11, and 12 at the centered $\Gamma$. **h** The surface local density of states for (001) surface along the high-symmetry directions. **i, j** The corresponding surface arcs at 3.65 and 3.55 THz, respectively. Even though BeAu exists in reality, we still found that around $\Gamma$ point one acoustic branch of BeAu shows the extremely small imaginary frequency, which cannot be removed in our current calculations, possibly due to misconsideration of long range interatomic interaction in the force constant construction or anharmonic effects.

detecting both single Weyl TPs and nontrivial long arc surface phonons by experiments.

**High degenerate Weyl TPs in noncentrosymmetric super-conductor BeAu.** High degenerate Weyl points refer to the nontrivial crossings, which have a degeneracy higher than two. Our screening has found the existence of threefold and fourfold nontrivial high degenerate Weyl points in 447 TP materials (see Supplementary Table 4). For threefold degenerate Weyl TPs, the Hamiltonian can be written as $\mathbf{H_3(k)} \propto \mathbf{q} \cdot \mathbf{S}$, where $\mathbf{q}$ is a wavevector and $\mathbf{S_i}$ are the rotation generators for spin-1 bosons. Those three bands at the crossing point will have the Chern numbers of $+2$, $0$, and $-2$ and they are also called "spin-1 Weyl point"[24]. For fourfold degenerate Weyl TPs, their Hamiltonian can be written as $H_4(\mathbf{k}) \sim I_2 \bigotimes (\mathbf{q} \cdot \boldsymbol{\sigma})$, where $I_2$ is the second-order identity matrix and $\sigma_i (i=x, y, z)$ are the three Pauli matrices. Those high degenerate Weyl TPs can be regarded as the sum of identical spin-$\frac{1}{2}$ Weyl points and they can be called "charge-2 Dirac point"[24] due to their Chern numbers.

Among 447 TP materials with high degenerate Weyl TPs, BeAu is a typical noncentrosymmetric superconductor with a critical temperature of 3.2 K (ref. [65]). BeAu crystallizes in a cubic B20 structure with the space group of $P2_13$ (no. 198). Both Be and Au atoms are located at $4a$ *Wyckoff* sites (Fig. 3a). The phonon spectrum along the high-symmetry lines in Fig. 3b shows six threefold degenerate Weyl TPs (blue circles at $\Gamma$) and six fourfold degenerate Weyl TPs (green circles at $R$). These high degenerate

Weyl TPs are protected by both the lattice symmetries (twofold screw rotations and threefold rotations) and TRS. To determine their topological natures, we calculate the Chern numbers for high degenerate Weyl TPs at both $\Gamma$ and $R$ in the BZ (Fig. 3c, d). The Chern numbers for threefold degenerate Weyl TPs are $-2$, $0$, and $2$, while those are $\pm 2$ for fourfold degenerate Weyl TPs. For the Weyl TPs at $\Gamma$, we can derive its Wannier center evolutions because of the well-separated bands. The threefold degenerate Weyl TPs in Fig. 3c are contributed from three phonon branche nos. 10, 11, and 12. As shown in Fig. 3f, the Wannier center evolutions for those three branches indicate that both nos. 10 and 12 bands are topologically nontrivial, whereas no. 11 band is trivial. This fact reveals that the threefold degenerate point is similar to a single Weyl point. However, it has a high topological charge of $+2$ and the corresponding Berry curvatures at $q_{xy}$ plane give the source behaviors at $\Gamma$ ($\omega = 3.669$ THz), which is protected by the twofold screw rotation axis at $\Gamma$ in the BZ. The fourfold degenerate Weyl TPs in Fig. 3d are contributed from two doubly degenerate bands, with the Chern numbers, $c = \pm 2$. These fourfold degenerate Weyl TPs at $R$ have the topological charge $-2$, which is also protected by the twofold screw rotation axis at $R$. Both the threefold and fourfold degenerate Weyl TPs have the opposite topological charges, indicating the conservation of the topological charges for Weyl TPs in the first BZ. Meanwhile, as shown in Fig. 3h, we have derived the topological nontrivial surface TPs along the high-symmetry lines on the (001) surface. Clearly, the topological nontrivial surface states connect both threefold and fourfold degenerate Weyl TPs at $\bar{\Gamma}$ and $\bar{M}$.

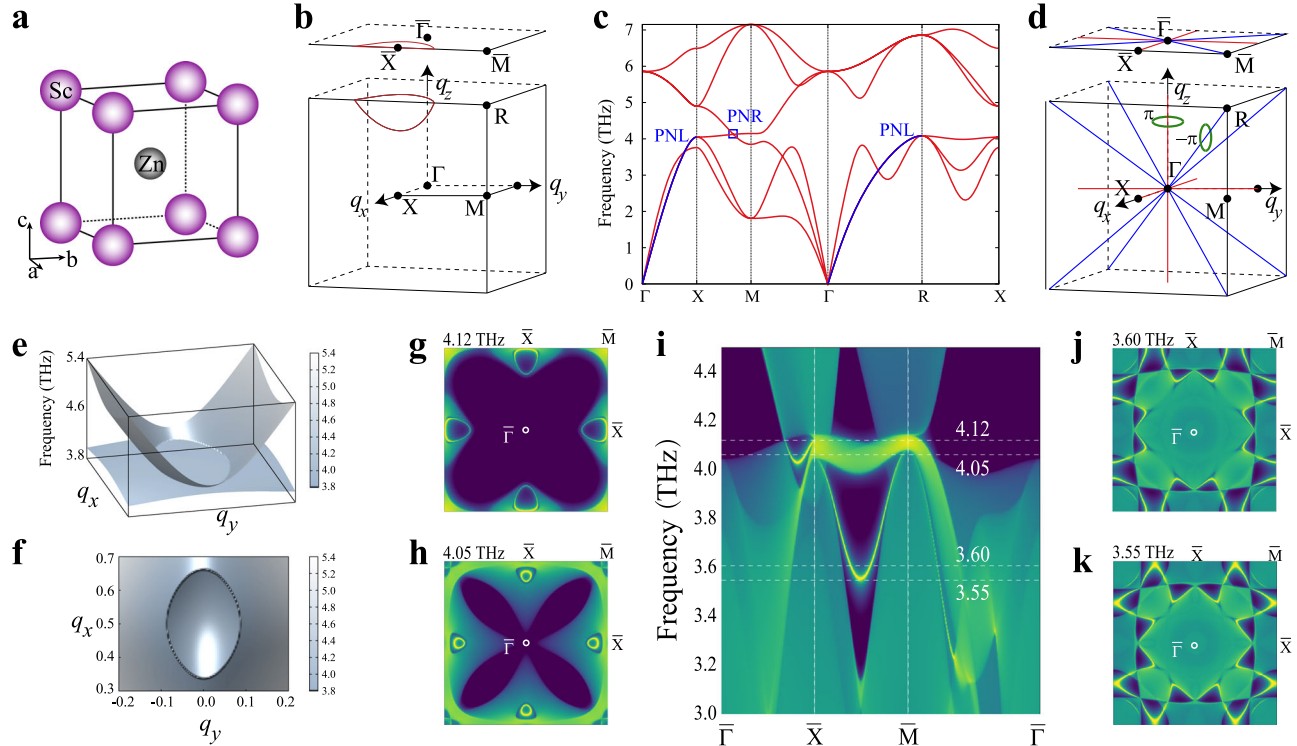

**Fig. 4 Phonon band structures and topological properties of phonon of ScZn. a** The crystal structure of ScZn (space group $Pm\bar{3}m$ 221). **b** The BZ of ScZn and the illustration of the nodal-ring TPs (red curve). **c** The phonon spectrum of ScZn. **d** The illustrations of the straight nodal-line TPs along the Γ–R and Γ–X lines in ScZn. **e**, **f** are the 3D phonon bands around the nodal-line TPs surrounding the M point. **g**, **h** The derived phononic surface states at the frequencies of 4.12 and 4.05 THz, respectively, of the (001) surface BZ (as defined in **b**). **i** The surface phononic spectrum of the (001) surface. **j**, **k** The phononic surface states at the frequencies of 3.60 and 3.55 THz, respectively.

Consequently, it can be clearly seen that two long arcs connect the projected double Weyl points at center-$\bar{\Gamma}$ and corner-$\bar{M}$, and those surface arcs are constrained by the TRS. The nontrivial open arc phononic surface states can be used to design the ideal negative refraction materials and this concept was recently realized in macroscopic metamaterials of Weyl phononic crystal[66].

**Nodal-line (ring) TPs in centrosymmetric ScZn.** Nodal-line (ring) TPs, formed by continuous phononic band crossings, have also been predicted in phonon spectra of materials[30–34]. Our results suggest that many materials exhibit such nodal-line (ring) TPs. Here, we introduce ScZn, with a B2 lattice structure (see Fig. 4a), that hosts both nodal-ring and straight line TPs as shown in Fig. 4c.

Firstly, we have noted that the phononic band crossing point between LA and TO on the X–M line of the BZ (Fig. 4c). Unlike the single Weyl TP in LiCaAs, this crossing is not an isolated point, but belongs to a closed ring formed by the continuous linear band crossings, as shown in Fig. 4e, f. The occurrence of the nodal-ring TPs in ScZn can be attributed to the presence of the $PT$ symmetry. Considering the mirror symmetry, ScZn totally hosts six nodal-ring TPs, which are located at the boundary planes of the BZ surrounding the M point in Fig. 4b. Secondly, the nodal-line TPs have been observed along the Γ–X and the Γ–R directions, as shown in Fig. 4c. They can be viewed as countless linear band crossings along the high-symmetry lines and extend through the whole BZ, as illustrated in Fig. 4d, similar to $MgB_2$ (ref. [31]). In particular, our further studies reveal that the nodal-ring TPs centered at the M point and the nodal-line TPs along the high-symmetry directions in ScZn are both protected by the $PT$ and mirror symmetries. For inversion symmetry P, we can

introduce the inversion operator as $\widehat{P} = \sigma_z$ to Eq. (1) to satisfy

$$H(\widehat{P}\mathbf{q}) = \widehat{P}H(\mathbf{q})\widehat{P}^{-1}. \tag{7}$$

We can simplify this equation into the following relations

$$\begin{aligned} d_{1,2}(-\mathbf{q}) &= -d_{1,2}(\mathbf{q}) \\ d_{0,3}(-\mathbf{q}) &= d_{0,3}(\mathbf{q}) \end{aligned} \tag{8}$$

which lead to that $d_{1,2}(\mathbf{q})$ are odd functions of $\mathbf{q}$, and $d_{0,3}(\mathbf{q})$ are even functions. Furthermore, the operator of the TRS can be written as $\kappa$ (a complex conjugate operator for the spinless case). It needs to mention that $H(\mathbf{q})$ and $T$ are to be commutable, $[H(\mathbf{q}), T] = 0$, which requires that

$$H(T\mathbf{q}) = TH(\mathbf{q})T^{-1} \tag{9}$$

Substituting $T$ with $\kappa$ leads to

$$\begin{aligned} d_{0,1,3}(-\mathbf{q}) &= d_{0,1,3}(\mathbf{q}) \\ d_2(-\mathbf{q}) &= -d_2(\mathbf{q}) \end{aligned} \tag{10}$$

where $d_{0,1,3}(\mathbf{q})$ are even functions and $d_2(\mathbf{q})$ is an odd function of $\mathbf{q}$. Due to the $PT$ symmetry from above two equations, we can obtain that $d_1(\mathbf{q}) = 0$, $d_2(\mathbf{q})$ is an odd function and $d_{0,3}(\mathbf{q})$ are even functions of $\mathbf{q}$. Ignoring the terms greater than the third power, $d_i(\mathbf{q})$ can be the followings,

$$\begin{aligned} d_2(\mathbf{q}) &= \sum_{i=x,y,z} \alpha_i q_i, \\ d_3(\mathbf{q}) &= b + \sum_{i=x,y,z} \beta_i q_i^2. \end{aligned} \tag{11}$$

At this stage, the eigenvalues of Eq. (2) can be simplified to $\omega(\mathbf{q}) = d_0 \pm \sqrt{d_2^2 + d_3^2}$ and the band crossing points

require $d_2(\mathbf{q}) = d_3(\mathbf{q}) = 0$. Here, we take the nodal-ring TPs centered at $M$ as an example to elaborate the role of the mirror symmetry. First, $d_2(\mathbf{q}) = \alpha_x q_x + \alpha_y q_y + \alpha_z q_z = 0$ can determine a plane passing the central point of the circle and the $d_3(\mathbf{q}) = b + \beta_x q_x^2 + \beta_y q_y^2 + \beta_z q_z^2 = 0$ is an equation of an ellipsoidal surface centered at $M$. The crossings between the plane and the ellipsoidal surface form a closed loop, which is the nodal ring centered at $M$. However, this nodal ring can tilt to arbitrary direction. Here, we can choose the mirror symmetry as $\widehat{M}_z = \sigma_z$ to give an additional constraint to the Hamiltonian,

$$H(\widehat{M}_z \mathbf{q}) = \widehat{M}_z H(\mathbf{q}) \widehat{M}_z^{-1} \quad (12)$$

Using $\sigma_z$ to replace $\widehat{M}_z$, we can obtain the following expressions constrained by mirror symmetry

$$
\begin{aligned}
d_{1,2}(q_x, q_y, q_z) &= -d_{1,2}(q_x, q_y, 1 - q_z), \\
d_{0,3}(q_x, q_y, q_z) &= -d_{0,3}(q_x, q_y, 1 - q_z).
\end{aligned}
\quad (13)
$$

Therefore, the $M$-centered nodal-ring TPs can be confined to the plane of $q_z = \pi$. For nodal-line TPs, the mirrors can also constrain them along one high-symmetry line. Since the nodal-line TPs possess nonzero Berry phases, there exist phononic nontrivial drumhead-like surface states. To elucidate this feature, we have calculated the phonon spectrum on the (001) surface with Green's function method in Fig. 4i. It shows the nontrivial surface states along the $\bar{\Gamma}$–$\bar{X}$ line and the $\bar{X}$–$\bar{M}$ line, respectively. When the bulk phonons are projected to the (001) surface, the nodal-ring TPs, perpendicular to the $q_z$ direction, can hold co-dimensions, which form a closed circle around the $\bar{X}$ point, as illustrated in Fig. 4b. Within the projected nodal-ring TPs, the nontrivial drumhead-like surface states form closed loops around the $\bar{X}$ point, which evolve with the phonon frequency (Fig. 4g with $\omega = 4.12$ THz and Fig. 4h with $\omega = 4.05$ THz). When the straight nodal-lines are projected onto the (001) surface, they are still along the high-symmetry paths to form the triangle region on the surface 2D BZ in Fig. 4d. Within this region, the nontrivial drumhead-like surface phononic states would occur. The nontrivial drumhead-like surface states induced by straight nodal lines are between 3.55 and 4.00 THz, and they connect two points on the neighboring projected straight nodal lines, as shown in Fig. 4j, k.

Our HT calculations further reveal that 114 CsCl-type materials isostructural to ScZn exhibit very similar phonon dispersions (see Supplementary Table 1). All of them host the nodal-line TPs within their acoustic branches along the $\Gamma$–$X$ and $\Gamma$–$M$ lines, but only some of them host nodal-ring TPs between the LA and TO branches. The existence of the nodal-ring TPs depends on the relative atomic masses of the constituents in compounds. If the atomic masses differ greatly, the nodal-ring TPs at zero acoustic-optical gap will disappear.

**Hourglass nodal-net TPs in centrosymmetric TeO₃.** Besides the above nodal-line (ring) TPs associated with the symmorphic symmetries, the nonsymmorphic space groups can produce the symmetry-enforced nodal-line (ring) TPs. Nonsymmorphic symmetries are formed by combining the operations of the point group $g$ and translations $\mathbf{t}$ by fraction. TeO₃ is such a case, which crystallizes in the *Pnna* space group (no. 52)[67]. In its unit cell, there are four Te atoms and each of them forms the TeO₆ octahedra (Fig. 5a). In this space group, six nonsymmorphic symmetries are present as follows: $g_1 = \{M_z | \mathbf{a}/2\}$, $g_2 = \{M_y | \mathbf{a}/2 + \mathbf{b}/2 + \mathbf{c}/2\}$, $g_3 = \{M_x | \mathbf{b}/2 + \mathbf{c}/2\}$, $g_4 = \{C_{2y} | \mathbf{a}/2 + \mathbf{b}/2 + \mathbf{c}/2\}$, $g_5 = \{C_{2z} | \mathbf{a}/2\}$, and $g_6 = \{C_{2x} | \mathbf{b}/2 + \mathbf{c}/2\}$. Our results reveal that TeO₃ possesses the HNN TPs in its phonon dispersion in Fig. 5b. To show the key

feature of the HNN TPs, here we focus on the four phonon branches (bands 25, 26, 27, and 28) with a frequency range from 12 to 14 THz. These four bands form two nodal-line TPs (marked in blue in Fig. 5b) along the boundaries of the BZ (Fig. 5c). The occurrence of those nodal-line TPs are protected by the crystal symmetries. By combining the TRS $T$ and $g_1 = \{M_z | \mathbf{a}/2\}$, the lattice momentum can be transformed to

$$
\begin{aligned}
g_1 &: (q_x, q_y, q_z) \rightarrow (q_x, q_y, -q_z), \\
T &: (q_x, q_y, -q_z) \rightarrow (-q_x, -q_y, q_z).
\end{aligned}
$$

This joint operation can be expressed as $\widetilde{T} = g_1 T$, which commutes with Hamiltonian $H(\mathbf{q})$ at $\mathbf{q} = (0, 0, q_z)$. Under the glide reflection operation $g_1$, for invariant lines or planes satisfying $g_1 \mathbf{q} = \mathbf{q}$, the Bloch states can be eigenstates $g_1 \left| u_{\mathbf{q}}^{\pm} \right\rangle = \pm \lambda e^{i\mathbf{q} \cdot \mathbf{t}} \left| u_{\mathbf{q}}^{\pm} \right\rangle$. Due to the spinless nature of phonons, $T^2 = 1$ cannot produce the Kramers-like degeneracy and $\lambda$ should be $\pm 1$. This means that the momentum $q$-dependent eigenvalues of $g_1$ are $\pm e^{iq_x/2}$. Under above conditions, $T^2 \equiv 1$ and $g_1^2 = e^{iq_x}$, the Kramers-like degeneracy($\widetilde{T}^2 = -1$) can be realized when $q_x = \pi$. It means that symmetry protected nodal lines should emerge along the high-symmetry $X$–$U$ ($\pi, 0, q_z$) and $S$–$R$ ($\pi, \pi, q_z$) lines. Similarly, $g_2 T$ and $g_3 T$ operations can enforce phononic band degeneracy (for the nodal-line occurrence) along the high-symmetry $Z$–$T$, $R$–$S$, $Z$–$U$, and $Y$–$S$ lines of the BZ (see Fig. 5c). The operation of the screw rotation $g_4$ guarantees the band degeneracy at any point on the $q_y = \pi$ plane, which implies the occurrence of the phononic nodal surface, as outlined in orange in Fig. 5c. In addition to the occurrence of phononic nodal line and nodal surface, we note that in three high-symmetry $\Gamma$–$X$, $Y$–$X$, and $U$–$R$ lines, the twofold degenerate phononic bands further split into the hourglass nodal points[68–70] (see blue and orange solid circles in Fig. 5b).

It needs to be emphasized that the hourglass nodal points (Fig. 5b) are protected by crystal symmetries. Here, we focus on the hourglass nodal point along the $U$–$R$ line in $q_x = \pi$ plane (*XURS*) to elucidate the role of nonsymmorphic symmetries. For the operation $g_3 = \{M_x | \mathbf{b}/2 + \mathbf{c}/2\}$, the eigenvalues are $g_{\pm}(q_y, q_z) = \pm e^{(iq_y/2 + iq_z/2)}$. Hence, at the point $U(\pi, 0, \pi)$, the degenerate phonon points have opposite eigenvalues ($\pm i$). Similarly, for the point $R(\pi, \pi, \pi)$, the eigenvalues would be $+1$ or $-1$. However, the $g_4 T$ operator protects the band degeneracy of the $S$–$R$ line and it commutes with $g_3$ at the $R$ point, which leads the same eigenvalues, ($+1, +1$) or ($-1, -1$) for the degenerate phonon bands at $R$. As a result, from $U$ to $R$, phonon bands have to switch partners and inevitably cross each other, which causes the occurrence of the hourglass phonon bands in Fig. 5d. Interestingly, this hourglass phonon nodal point is not an isolated one and there exists an hourglass nodal ring on the *XURS* plane. As illustrated by the black curve starting from the $S$ point in Fig. 5c, the hourglass nodal point along the $U$–$R$ line is a point on this ring. To clearly visualize the shape of the hourglass phonon nodal ring, we plot the two crossing phononic bands on the *XURS* plane in Fig. 5e. Because the start and end points of this nodal ring are at the same point $S$ in the corner of BZ, this hourglass nodal ring can form the hourglass nodal chain (HNC; see the blue curve in Fig. 5g), which goes through the whole BZ along the $q_z$ direction. Similarly, the HNC can also be observed on the $q_z = 0$ plane ($\Gamma XSY$). For the glide reflection operation $g_1$, the eigenvalues of $g_1$ are $\pm e^{iq_x/2}$. The eigenvalues at $X(\pi, 0, 0)$ and $Y(0, \pi, 0)$ can be identified to $\pm i$ and $\pm 1$, respectively. Since the twofold screw axis $g_4$ commutes with $g_1$, the eigenvalues at $Y$ can be confirmed to be ($+1, +1$) or ($-1, -1$), while the $X$ point holds the opposite eigenvalues ($+i, -i$), as shown in Fig. 5h. Hence, the

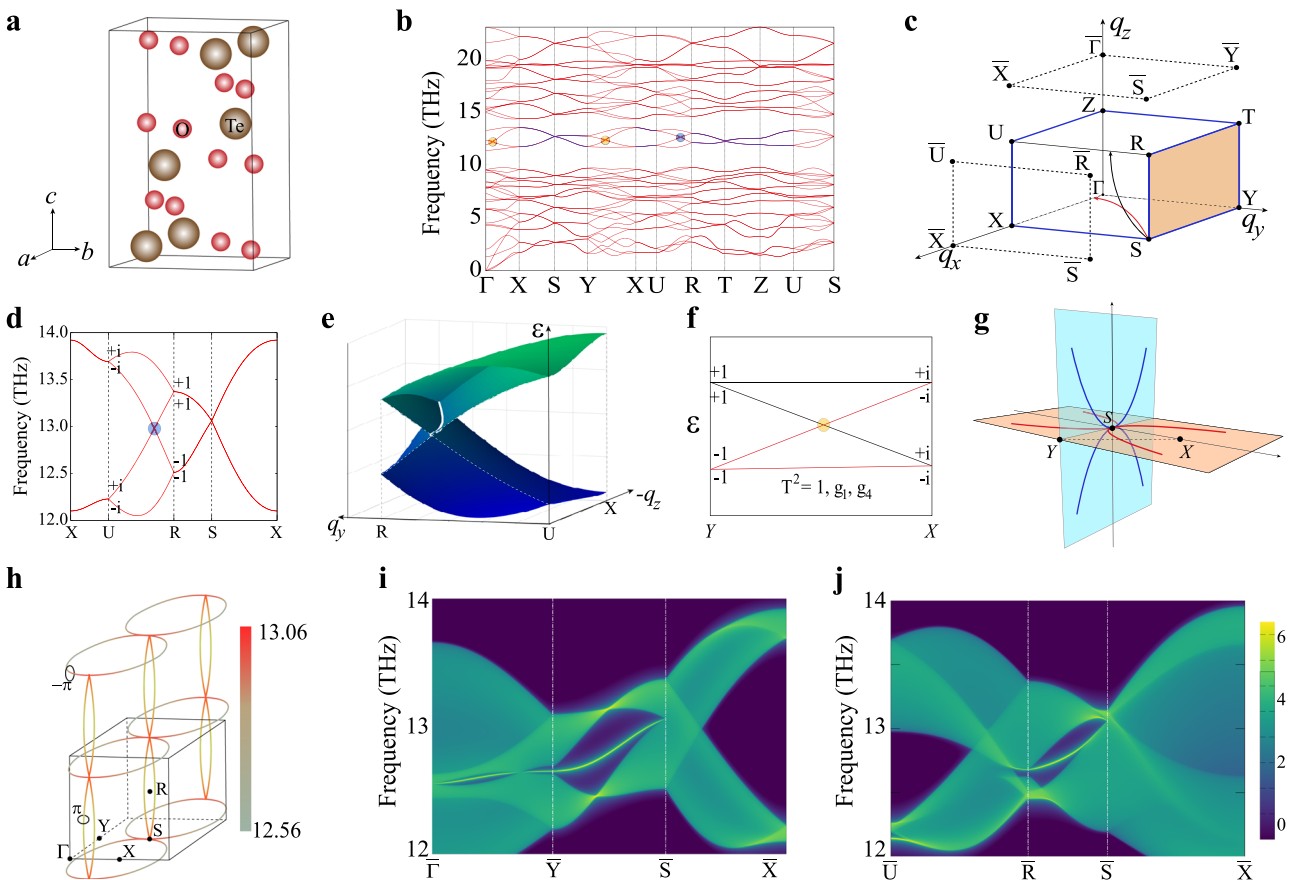

**Fig. 5 Hourglass nodal-net TPs of TeO₃. a** The unit cell of TeO₃ (space group *Pnna* 52). **b** The phonon dispersion of TeO₃. **c** The bulk BZ and the (001) and (100) surface BZs. Both the red and black arrows denote the hourglass nodal-line TPs, whereas the blue lines along the BZ boundary represent the twofold degenerate nodal-line TPs. All points on the $q_y = \pi$ plane are twofold degenerate nodal points (called twofold degenerate nodal surface), as shown in the yellow plane. **d** The phonon spectrum along *X–U–R–S–X* at the $q_{yz}$ plane. The eigenvalues of $\{M_x|\mathbf{b}/2 + \mathbf{c}/2\}$ are given at both *U* and *R* points. **e** The 3D phonon dispersions at the $q_{yz}$ plane. The solid white line represents the hourglass nodal line on which any point represents a hourglass nodal point. **f** The phononic hourglass nodal dispersions protected by $g_1$ and $g_4$. The eigenvalues of $\{M_z|\mathbf{a}/2\}$ are given at both *Y* and *X* points in the BZ. **g** Hourglass nodal chains (HNCs) at the $q_{xy}$ plane (red solid lines) and at the $q_{yz}$ plane (blue solid lines). Both these HNCs connect to each other only at the *S* point to form an hourglass nodal net (HNN) in the extended BZ. **h** The shape of the HNN in the three-dimensional view. Two black circles are used to calculate the Berry phase of the HNCs and the color represents the energy dispersion of the HNN in THz. **i, j** The phononic surface states along the high-symmetry lines for the (001) and (100) surfaces, respectively.

hourglass nodal point appears along the *X–Y* line. On the Γ*XSY* plane, the similar hourglass nodal points appear to form a closed ring also starting from the *S* point (see the red curve in Fig. 5c). Thus, the HNC can also be observed in the extended BZ (see the red curve in Fig. 5g), which also goes through the whole BZ on the $q_z = 0$ plane. In particular, both HNCs originating from *S* are perpendicular to each other which form the HNN as shown in Fig. 5g, h.

Furthermore, the topological nontrivial surface states have been calculated on the (001) and (100) surfaces. Due to the co-dimension rule[71], the surface states of the HNC on $q_z = 0$ plane can be obtained on the (001) surface, whereas the surface states of the HNC on the $q_x = \pi$ plane emerge on the (100) surface. As the feature of nodal rings, the drumhead-like surface states can be observed inside or outside the projected region of HNCs, which are determined by the region with nonzero Berry phase[32]. As shown in Fig. 5i, j, the nontrivial surface states occurs outside the projected HNC on the (001) surface (Fig. 5i), while the drumhead-like surface states appear inside the HNC on the (100) surface (Fig. 5j). HNNs are protected by the nonsymmorphic symmetry and can be extremely stable. Due to the diversity of crystal symmetries, more interesting topological phonon nodal lines or rings can be expected.

## Discussions

From our HT analysis, we have systematically classified those TP materials into two main categories of Weyl TPs and nodal-line (ring) TPs. Nodal-line (ring) TPs accounted for the largest number of TPs. Among 5014 TP materials, 4978 materials exhibit nodal-line (ring) TPs (in Supplementary Tables 3–5). Figure 6 shows the schematic relations between different types of TPs. As shown in Fig. 6c, nodal-ring TPs can be visualized as continuous crossings of two phononic bands forming a closed loop. Non-trivial drumhead-like surface states, the key feature of topological nature, can be observed when the bulk nodal-ring TPs are pro-jected onto a given surface. These nodal-line (ring) TPs can further evolve into various nodal-link, nodal-chain, or nodal-net TPs, if some crystal symmetries (e.g., mirror or screw symme-tries) are combined with the *PT* symmetry. Here, we elucidate the evolution of cases of the nodal-link and nodal-net TPs. Once the mirror symmetry (*M*) is added into the *PT* symmetry, it is pos-sible to obtain various nodal-link TPs. When the crystal hosts two noncoplanar nodal rings protected by two mirrors, two perpen-dicular nodal rings can pass through the inside of each other and form the Hopf-link TPs by hooking each other, as shown in Fig. 6d. The Hopf-link semimetal states of Fermions have been predicted in Co₂MnGa (ref. [72]) and the phononic counterpart

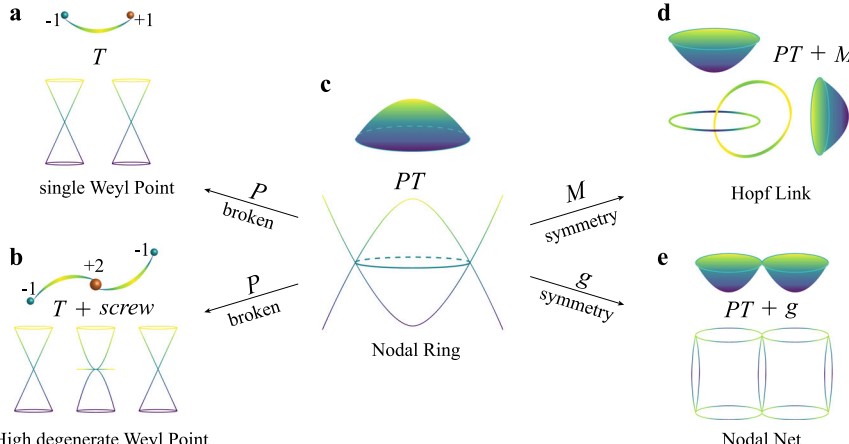

**Fig. 6 The schematic relations between different types of TPs. a** Single Weyl TPs in the open arc states. **b** High degenerate Weyl TPs in the arc surface states. **c** Nodal-ring (line) TPs in drumhead-like surface states. **d** Hopf-link TPs in nontrivial surface states. **e** Nodal-net TPs (e.g., HNN) and their corresponding drumhead-like surface states.

can also be realized. Once the nonsymmorphic symmetry (*g*) is combined with the *PT* symmetry, it is possible to obtain various nodal-net TPs, comprised by continuous nodal-line (ring) TPs. As discussed in $TeO_3$, the nonsymmorphic symmetry (*g*) can induce the hourglass nodal points, which can form the nodal chain in the extended BZ. Those HNCs share the same vertex with the one in the vertical plane and the HNN can be obtained as shown in Fig. 6e. Currently, we have identified 15 HNN TP materials (Supplementary Table 6). HNN and Hopf-link TPs still exhibit the drumhead-like surface states that are similar to nodal-ring TPs (Fig. 6d, e). Moreover, due to the diversity of both *g* and *M* symmetries, more interesting and nontrivial TPs forming by nodal-line (ring) TPs can be discovered.

Furthermore, these nodal-line (ring) TPs can be split into Weyl TPs once its *P* symmetry is broken. Considering the degenerate degree of the nontrivial crossings, Weyl TPs can be categorized into single Weyl TPs (see Fig. 6a) and high degenerate Weyl TPs (see Fig. 6b). The single Weyl TPs can appear at arbitrary momentum positions, while the high degenerate Weyl TPs must stay at the high-symmetry points protected by the screw symmetry. When the single Weyl TPs are projected onto the surface, one open arc will connect those TPs with opposite charge, as shown in Fig. 6a. Due to the high topological charge feature of high degenerate Weyl TPs, multiple open arcs can be observed, which originate from the projected Weyl nodes, as shown in Fig. 6b. For instance, in BeAu, the projected Weyl node of the threefold degenerate Weyl TPs is connected by two open arc states. Interestingly, our calculations predicted 36 materials only for single Weyl TPs (Supplementary Table 2), 463 materials for mixed single Weyl TPs and nodal-line (ring) TPs, and 266 materials for mixed single Weyl, high degenerate Weyl and nodal-line (ring) TPs (Supplementary Table 4), and 181 materials for mixed high degenerate Weyl and nodal-ling (ring) TPs.

When a large set of TP materials data is available, one can expect a variety of new phenomena by manipulating chemistry and structure. Therefore, the researches on TPs certainly deserve to be exploited further.

First, those *TP* materials can be modulated by the pseudospins[36,73] and pseudo-SOC[74,75] from the crystalline symmetry and pseudoangular momenta. Phonon Hall effect[62,76,77] has been experimentally observed in a paramagnetic insulator, which can be applied to ballistic thermal transport[35] and Berry-phase-induced heat pumping[78]. When the *P* symmetry is broken, a pair of valley polarized boundary states with a locked valley-momentum can lead to the phonon quantum valley Hall

effect[36], with potential applications as the phonon valley filter[79], phononic antennas[80], and negative refractive index materials[66]. When the TRS is broken, the phonon quantum anomaly Hall effect (QAHE) can be realized by introduction of ionic lattices by the Lorentz force on charge ions[35,81], magnetic lattices by the Raman-type spin–lattice interaction[35,62], or a Coriolis/magnetic field[46,82]. The phonon QAHE hosts the one-way edge states which are immune to scattering from defects, and this unique state can be used for novel phonon devices, such as phonon diodes and waveguides[38,46,83,84], acoustic delay lines[85], thermal rectification[44,86–88], and dissipationless phononic circuits[38,89,90].

Second, TPs can also enhance the thermoelectric properties of materials. The thermoelectric performance is determined by the thermal power of merit, $zT = \frac{S^2\sigma}{\kappa_e + \kappa_l}$, where $S$ is the Seebeck coefficient, $\sigma$ is the electrical conductivity, and $\kappa_e$ and $\kappa_l$ are the electronic and phononic contributions to thermal conductivity. To improve $zT$, we can either decrease the overall $\kappa_e/\kappa_l$ values or increase $S$. For an undoped material, it is difficult to reduce $\kappa_e$ since it is related to $\sigma$ according to the Wiedemann–Franz law[91]. Therefore, a more realistic approach is to decrease $\kappa_l$ caused by phonons traveling through the lattice. Different from the non-TP materials, the gapless topological phonon modes in the TP materials can provide more scattering channels in the three phonon–phonon scattering processes to decrease the mean free path and suppress the $\kappa_l$ (ref. [41]). Furthermore, it is still likely that the TPs and topological electrons can coexist in the same material, because the topological properties are governed by the crystal symmetry[23,33,92,93]. Since the enhancement of electronic DOS can increase $S$, one can design the topological electronic materials with nodal points near Fermi level to optimize the figure of merit in combination with TPs for suppressing the $\kappa_l$. In addition, the clean TPs' surface states can enhance the electron–phonon coupling and possibly trigger the topological superconductivity at the surfaces or interfaces[21,93,94].

Third, TPs can be physically detected at the whole phonon spectrum by using the techniques, such as infrared spectroscopy[95], x-ray scattering[96], Raman techniques[97], and inelastic neutron scattering[98,99]. Recently, two inelastic x-ray scattering studies have been applied to detect the topological phonons in FeSi and $MoB_2$ (refs. [34,40]). However, it is far more challenging to characterize the surface phonons because X rays have only a penetration depth of the micron scale. For the surface phonons, several techniques, such as helium scattering[100], terahertz polarimetry[101,102], and high-resolution electron energy loss spectroscopy (EELS)[103,104], can be applied. In particular, the

lattice vibrations of Be(10$\bar{1}$0) and Be(0001) surfaces have been successfully measured from EELS[105,106]. While the experimental characterization of surface phonons for a single material remains challenging, our current work can undoubtedly facilitate this research by providing over 300 ideal candidate materials for experimental verification.

In summary, we have developed a HT and data-driven approach to evaluate the TPs in over 10,000 materials, using the existing phononic database and our in-house calculations. Our screening suggests that TP states are universally present, highlighting extensive possibilities for realizing TPs in a variety of materials toward different potential applications. We expect more topological bulk phonons and nontrivial edge states to be detected by experiments in near future. As such, many exciting phenomena, such as topological superconductivity and high thermoelectricity, may be realized by utilizing the topological phonons in those identified TP materials.

## Methods

**Phonon calculation**. All DFT calculations have been performed by Vienna ab initio simulation package, based on the projector augmented wave potentials and the generalized gradient approximation within the Perdew-Burke-Ernzerhof for the exchange correlation treatment. The force constants downloaded from the PHONONPY database were calculated by the finite displacement method. For the in-house phonon calculations, we used the density function perturbation theory. We performed the geometry optimization of the lattice constants by minimizing the forces within 0.001 eV/Å. The cutoff energy for the expansion of the wave function into the plane waves was set to 1.5 times of the ENMAX in the POTCAR. For the topological analysis, we used the conjugate gradient method in SciPy[107] to get the crossings, and calculated the Berry phase and Chern number to identify the nontrivial topological natures. To determine the topological charge, the Wilson-loop method[108,109] was chosen. A sphere centered at a WP was sliced into independent orbitals by a constant polar angle $\theta$ and the evolution of Wannier centers (phase factor $\phi$) on orbitals can give topological charges of WPs. For the surface DOS, we used force constants as tight-binding parameters to construct surface and bulk Green's functions, and the imaginary part of the Green's function produces the DOS[110,111].

## Data availability

We have provided a Supplementary Material including 14,248 pages and 5014 figures to classify all 5014 TP materials according to the geometrical character and the Berry phases of topological nodal points (Weyl node, Dirac node, and high degenerate nodal points) and nodal-line (ring) TPs. Each material entry includes the spatial information of the points (e.g., $x$, $y$, $z$ coordinates, frequencies, modes, and band paths), and multiplicity, degeneracy, and topological charges to each phononic band crossing points. These data, together with the interactive visualization of atomic structures and phonon band dispersion, are also available, if requested. In addition, we have constucted the corresponding online database, available at www.phonon.synl.ac.cn or https://tpdb.physics.unlv.edu/.

## Code availability

In order to effectively and conveniently analyze topology of phonons, we developed an HT-TPHONON code to automate all processes (as shown in Fig. 1) and connect them with the DFT calculations based on Python scripting. All codes used in this work are either publicly available or available from the authors upon reasonable request.

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

## Acknowledgements
Work at IMR was supported by the National Science Fund for Distinguished Young Scholars (grant number 51725103), by the National Natural Science Foundation of China (grant number 51671193), by the Science Challenging Project (grant number TZ2016004), and by major research project 2018ZX06002004. Work at UNLV is supported by Q.Z.'s startup grant. All calculations have been performed on the high-performance computational cluster in Shenyang National Park and XSEDE (TG-DMR180040).

## Author contributions
X.-Q.C. proposed this idea, and both X.-Q.C. and Q.Z. designed the research. J.X. Li, J.X. Liu, M.F.L., L.W., R.H.L., Y.C., D.Z.L., Q.Z., and X.-Q.C. performed and analyzed the calculations and contributed to interpretation and discussion of the data. S.A.B., Q.Z., J.X. Li and X.-Q.C. coded the online topological phonon database. X.-Q.C., J.X. Li, and Q.Z. wrote the manuscript. All authors discussed this manuscripts.

## Competing interests
The authors declare no competing interests.
