## [Peer Review File · Nature Communications]

REVIEWER COMMENTS

Reviewer #1 (Remarks to the Author):

The authors in the paper evaluate gapless topological phenomena including Weyl nodes, high degenerate Weyl nodes and nodal-line (ring) in topological phononics among over 10000 materials. They also identify 322 TP materials with clean nontrivial surface states. I think that the results are important given that it provides a library of TP materials. But I have a concern that their evaluation method looks straightforward and does not substantially advance our understanding of topology. Moreover, it looks that their classification does not predict any novel gapless topological phases. With these concerns, I cannot recommend it for publication in Nature Communication, at least at this stage. I have also some questions and comments in the following.

1. Is the library obtained in the paper complete?
2. In the last sentence of paragraph 2 of page 1, the authors state "TPs would be very promising for applications in the abnormal heat transport, solid-state refrigeration, and phonon wave-guides, and so on". The paper would read better if the authors can provide some relevant references for these applications.
3. In paragraph 5 of page 2, the authors write "After optimization, the identified inversion points ...". I wonder what "inversion points" refer to. Also in this paragraph, the authors state "we also checked if the points are at or off the high-symmetry paths". I wonder why it should be checked in this step.
4. In the following paragraph, the authors argue that Weyl TPs cannot exist in a system with PT symmetry. There are two cases for PT symmetry: $(PT)^2 = -1$ and $(PT)^2 = 1$. In the former case, the energy spectra are always two-fold degenerate, usually leading to Dirac nodes if gapless points exist. Do the authors here only consider the latter case? What about the former one?
5. In the same paragraph, the authors mention "we only looked for nodal-line in materials with PT symmetries". I wonder whether the authors only consider the nodal lines and rings protected by PT symmetry. If that is the case, I suggest putting this discussion in step (2).
6. In paragraph 3 of page 3, the authors write that the nodal-ring and nodal straight line TPs are protected by the mirror symmetry. Can the authors elaborate on it?
7. The authors calculate the Chern number over a closed surface to identify a Weyl point. I wonder how the Chern number is numerically evaluated given that the wave functions usually suffer a global phase uncertainty.
8. In Fig. 2f, the authors plot the Wannier center evolution. In a torus, one can calculate the Wannier centers along one direction with respect to another momentum. Can the authors briefly explain how this is performed in a closed surface like a sphere? What is theta?
9. Other minor problems: "exists" in "there exists two kinds of high ..." should be "exist". "The" in "For instance, The single type-I Weyl ..." should be "the".

Reviewer #2 (Remarks to the Author):

The authors did high-throughput searching topological states on phonon systems. Although the data screening took a lot of time, I think this paper is more suitable for Nature scientific data instead of Nature communication due to its lack of innovation. The new predicted materials are also not quite interesting to this community. So I don't recommend it to be published in Nature communication. By the way, there is another question about the workflow shown in Fig1. How do you calculate the Chern number when you have nodal line? I mean the step between "P-T inversion symmetry?" to "To compute Chern number".

Reviewer #3 (Remarks to the Author):

The authors have presented a systematic investigation and algorithm for the search of topological phononic states in solids with over 10^4 materials with 5014 topological phononic materials identified. Some important classes and examples are elaborated. This work is impressive. However, I have several comments:

1. I basically think the manuscript is too technical which narrows the scope of audience. The authors could write in a more transparent, physical and materials way, trying to state the findings in plain languages and avoid using jargons as much as possible.
2. The examples are discussed too briefly (Nature Commun. allows much more space than the length of the current manuscript), making the main body of the manuscript less attractive.
3. Most importantly, for the examples analyzed by the authors, the generic underlying physics is unclear especially those which are unique to phonons.

I strongly suggest the following revisions:

1. Possible physical properties of topological phonons, e.g., optical or transport properties, must be discussed in as much details as the author can provide.
2. Possible detections of topological phonons must be discussed.
3. Topological phonon properties are also discussed in mechanical metamaterials, e.g., Nature volume 555, pages342–345(2018); Phys. Rev. B 97, 180101(R), These works should be mentioned.

In addition, Refs. 33 and 46 are redundant.

Response to Referees' report
(manuscript NCOMMS-20-23399)

Detailed responses to Referee #1:

Comment #1.1: *The authors in the paper evaluate gapless topological phenomena including Weyl nodes, high degenerate Weyl nodes and nodal-line (ring) in topological phononics among over 10000 materials. They also identify 322 TP materials with clean nontrivial surface states. I think that the results are important given that it provides a library of TP materials. But I have a concern that their evaluation method looks straightforward and does not substantially advance our understanding of topology. Moreover, it looks that their classification does not predict any novel gapless topological phases. With these concerns, I cannot recommend it for publication in Nature Communication, at least at this stage. I have also some questions and comments in the following.*

Authors' Reply: We thank the referee very much for his/her nice summaries, suggestions and comments of our manuscript. Topological phonons were predicted in solid materials for the first time in 2018, which pointed out the double-Weyl phonon in transition-metal monosilicides [1] and single Weyl phonons in WC-type materials [2]. After that, topological phonons have attracted much attention due to the underlying relations to the dynamical process such as heat conduction, superconductivity, and electron-phonon coupling [3] within THz phonons. To date, the topological phonons have been studied in only a few materials that hold topological gapless phonons with double, triple or quadruple degeneracy. Compared to electrons, phonons have many different features in motion equations, orbitals, spin and statistics, and electromagnetic interaction. Despite above differences, both topological phonons and fermions are elementary excitations, whose energy dispersion can be generally written as $H(\mathbf{k}) = \sum_{i,j} k_i A_{ij} \sigma_{ij}$, where \mathbf{k} is the wave vector in reciprocal space, A is a matrix of coefficients, σ_0 is the 2×2 unit matrix and $\sigma_j, j = x, y, z$ are the three Pauli matrices [4]. So they largely share the same topological concepts, such as Berry phase, Chern number, Berry curvature. In all previous publications of topological phonons, Berry phase and Chern number are usually used as the topological invariants to characterize topologically nontrivial states. For topological electronic [5-7], people also used symmetry indicators [8, 9], band combinatorics [10] and topological quantum chemistry (TQC) [11]. Those methods can be extended to topological phonon as shown in two recent publications [12, 13].

The reasons why we choose the straightforward method to calculate the Berry phase and Chern number to evaluate gapless topological phonons are the followings,

- a) The phonon obeys the Bose-Einstein statistics and there is no concept of Fermi level. We need to analyze the phonon dispersion of all frequency ranges, instead of only considering the electrons near the Fermi level. This fact will bring much

heavier calculations for our work flow than all previous method including symmetry indicators [8, 9], band combinatorics [10] and topological quantum chemistry (TQC) [11].

- b) In two recent publications based on symmetry analysis [12, 13] for topological phonons, the authors only focused on the Weyl points at the high-symmetry points, which could not supply a comprehensive and feasible strategy for topological phonons classifications.
- c) Without the external field, the intrinsic topological phonons of solid materials are gapless phases due to the time-reversal symmetry. We focus on the gapless phases searching in this research. It is the most feasible and efficient method to use the Berry phase or Chern number to evaluate the topological nature of those crossings.

The true innovation in our manuscript is the realization of the high-throughput, robust and effective workflow to seek topological phonons for all possible phononic bands based topological theory. This is definitely not a simple task.

In this revised manuscript, the more important thing is to advance our understanding of topological phonons from the following aspects,

- a) Topological phonons extensively exist in phonon spectra of many known materials, which can be classified into two basic types of Weyl and nodal-line (ring) topological phonons. Weyl phonons can be classified into single and high degenerate Weyl phonons. The former is always associated with the inversion symmetry broken, whereas the latter still requires the screw axis. Nodal-line (ring) phonons can be grouped into four subgroups of individual nodal-line (ring), nodal-link, nodal-chain and nodal-net phonons, upon the manipulation of the PT symmetry and the nonsymmorphic symmetry (e.g. screw rotation or glide reflection or both). We have elucidated the physical mechanism for the occurrence of Weyl and nodal-line (ring) topological phonons. Furthermore, we have emphasized their mutual relationship among various types of topological phonons in new Fig. 6.
- b) For the first time we have predicted the interesting hourglass nodal net topological phonons in TeO_3 (new Fig. 5). This kind of topological hourglass nodal net phonons is very special, which is highly robust due to the nonsymmorphic symmetry protection. The hourglass nodal net phonons can induce the topologically protected drumhead-like surface states. Due to the diversity of nonsymmorphic symmetry, more novel topological phonons can be indeed expected.
- c) For the first time we also found the clean and single type-I Weyl phonons formed by the exact touching in the phononic acoustic-optical gap in half-Hesuler

LiCaAs (Fig. 2). Physically, this existence of Weyl point associated with the zero acoustic-optical gap may significantly enhance the phononic scattering, as compared with the non-zero acoustic-optical gap phonons. This fact would have potential effect in reducing crystalline thermal conductivity.

- d) Last, our results reveal that the extensive coexistence of different types of topological phonons in materials, such as, the coexisted three-fold and four-fold degenerate Weyl phonons in unconventional superconductor of BeAu and the coexisted nodal-line and nodal-ring topological phonons in ScZn. The nonsym-morphic symmetry plays an important role for the formation of the high degenerate Weyl phonons and nodal-net topological phonons.

Therefore, the key achievements of this work are (1) to provide a comprehensive identification and classification of topological gapless phonons; (2) to discover novel topological phonons which are feasible to detect in real materials; (3) to provide a library of structural motifs to facilitate the design of topological phononic metamaterials.

Comment #1.2: *Is the library obtained in the paper complete?*

Authors' Reply: In our current work, we have evaluated 13000 materials and constructed a topological phonon database of 5014 materials. For the first stage, the 5014 topological materials can cover all kinds different topological phonons and the library is complete for those 5014 topological materials. We are still performing the fully automated work flow on our supercomputing center of 15000 structures with experiments data and this will last for another 2 years due to the expensive calculations of force constants.

Comment #1.3: *In the last sentence of paragraph 2 of page 1, the authors state "TPs would be very promising for applications in the abnormal heat transport, solid-state refrigeration, and phonon wave-guides, and so on". The paper would read better if the authors can provide some relevant references for these applications.*

Authors' Reply: Thanks for your suggestions and we are sorry to miss those import references. For topological phonon, it's an import research direction to analyze the connections between the topological nature and application. At the current stage, the majority of studies are still limited to theoretical predictions and experimental verification of materials with the presence of novel topological phonons. Indeed, the TP materials are believed to be promising for applications such as heat transport and superconductivity. However, one needs to computationally solve the Boltzmann transport equation or the Eliashberg coupling function to evaluate the feasibility for a concrete material. As a starting point, we have been working on the these directions [3] and we wish this work of topological phonon database will encourage more studies in both theory and experiment. We have added the relevant references in the main text. In addition, we

wrote a new paragraph to explain the potential applications in detail.

Comment #1.4: *In paragraph 5 of page 2, the authors write “After optimization, the identified inversion points ...”. I wonder what “inversion points” refer to. Also in this paragraph, the authors state “we also checked if the points are at or off the high-symmetry paths”. I wonder why it should be checked in this step.*

Authors’ Reply: Thanks for your suggestions and the “inversion points” is ambiguous in here. The “inversion points” actually refer to the crossings with nonzero Berry phase and it’s used to describe the nontrivial points with band inversion in topological electrons. In phonon system, as illustrated in Ref [2], from the perspective of band order, the inversion points can be understood by the band inversion at the two ends (high symmetry points) of a high symmetry line and those crossing bands have disparate vibration modes. To avoid this confusion, we have used the “nontrivial crossings” to replace the “inversion points” in paragraph 5 and paragraph 6 of page 2. There are two reasons for us to check if the points are on or off the high-symmetry: 1) we stored points with this indicator (on or off high-symmetry line) to determine whether they should be used when we plot the phonon dispersion along high symmetry line. 2) The nontrivial crossings located at high symmetry lines usually have a cleaner environment and it’s beneficial to be observed on the surface. So when we choose the ideal clean topological phonon materials, we also consider this factor.

Comment #1.5: *In the following paragraph, the authors argue that Weyl TPs cannot exist in a system with PT symmetry. There are two cases for PT symmetry: $(PT)^2 = -1$ and $(PT)^2 = 1$. In the former case, the energy spectra are always two-fold degenerate, usually leading to Dirac nodes if gapless points exist. Do the authors here only consider the latter case? What about the former one?*

Authors’ Reply: Thank you much for your concerns on the PT symmetry. Due to their spinless nature of phonons, the Kramers degeneracy is always absent when the time-reversal symmetry (TRS) is preserved. So we only consider the latter case of $(PT)^2 = +1$ for real materials, as elaborated in our current manuscript. Indeed, in phonon system for intrinsic materials there is no such a case with PT symmetry of $(PT)^2 = -1$.

Comment #1.6: *In the same paragraph, the authors mention “we only looked for nodal-line in materials with PT symmetries”. I wonder whether the authors only consider the nodal lines and rings protected by PT symmetry. If that is the case, I suggest putting this discussion in step (2).*

Authors’ Reply: Thank the referee very much for this kind suggestion. The nodal line and nodal ring occur not only in materials with PT symmetries, but also in materials without inversion symmetries. As for point band touching, imposition of PT symmetries is too restrictive for the occurrence of nodal line (ring) and they can also exist under

the protection of discrete symmetries [16]. In step (2), we aim to identify the topological nodal line (TNL) along high symmetry line because we need not consider the PT symmetry for TNL. And we utilize two features of them, the nontrivial Berry phase and the continuous points along high symmetry line, to simplify our workflow and reduce our target data points due to the extensive existence of TNLs. TNL(R)s occur either along high symmetry line or off high symmetry line [15]. In case of missing any possible TNL(R)s off high symmetry line, we have added the step (4). When the PT symmetry is present, no Weyl point is possible and we seek all possible nodal points with the nontrivial Berry phases which definitely belong to TNL(R)s off high symmetry line. With the PT symmetry broken, we have considered all possible existence of Weyl points and TNL(R)s through computing Chern number.

Thus, our original sentence “we only looked for nodal-line in materials with PT symmetries” in our manuscript wasn’t written clearly enough and we changed it as follows, “As a result, when the PT symmetry is present, we just need to seek nodal-lines (rings) off high symmetry line.”

In addition, we have also added more details for our classifications to avoid unnecessary misunderstanding and to enhance the readability of our manuscript.

Comment #1.7: *In paragraph 3 of page 3, the authors write that the nodal-ring and nodal straight line TPs are protected by the mirror symmetry. Can the authors elaborate on it?*

Authors’ Reply: Thanks for this important suggestion. To elaborate this point, we first write a general Hamiltonian $H(\mathbf{q}) = \sum_{i=0}^3 d_i(\mathbf{q})\sigma_i$, in which σ_0 is a 2×2 identity matrix and $\sigma_i = x, y, z$ denote the Pauli matrices, respectively, $d_i(\mathbf{q})$ are real functions, and $\mathbf{q} = (q_x, q_y, q_z)$ are wave vectors of phonons.

For ScZn, its space group is $Pm\bar{3}m$, and we can choose the inversion operator as $\hat{P} = \sigma_z$. The inversion symmetry constrains the Hamiltonian, satisfying

$$H(\hat{P}\mathbf{q}) = \hat{P}H(\mathbf{q})\hat{P}^{-1}.$$

We can simplify this equation into

$$d_{1,2}(-\mathbf{q}) = -d_{1,2}(\mathbf{q}), \quad d_{0,3}(-\mathbf{q}) = d_{0,3}(\mathbf{q}).$$

This leads to $d_{1,2}(\mathbf{q})$ being odd function of \mathbf{q} and $d_{0,3}(\mathbf{q})$ are even functions. In addition, the time-reversal symmetry (T) is always preserved because the spinless nature of phonons, and the T symmetry can be κ (a complex conjugate operator for the spinless case). It needs to mention that $H(\mathbf{q})$ and \mathcal{T} are to commute, $[H(\mathbf{q}), T] = 0$, which requires that

$$H(T\mathbf{q}) = TH\mathbf{q}T^{-1}.$$

Substituting \mathcal{T} with κ leads to

$$d_{0,1,3}(-\mathbf{q}) = d_{0,1,3}(\mathbf{q}), \quad d_2(-\mathbf{q}) = -d_2(\mathbf{q}).$$

This leads to $d_{0,1,3}(\mathbf{q})$ being even functions and $d_2(\mathbf{q})$ is odd function of \mathbf{q} . Combining the constraints of time-reversal and space inversion symmetry from above two equations, we can obtain that $d_1(\mathbf{q}) = 0$, $d_2(\mathbf{q})$ is an odd function and $d_{0,3}(\mathbf{q})$ are even functions of \mathbf{q} . Ignoring items greater than the third power, $d_i(\mathbf{q})$ can be the followings:

$$\begin{aligned} d_1(\mathbf{q}) &= 0, \\ d_2(\mathbf{q}) &= \sum_{i=x,y,z} \alpha_i q_i, \\ d_3(\mathbf{q}) &= b + \sum_{i=x,y,z} \beta_i q_i^2. \end{aligned}$$

At this stage, the eigenvalues are $\omega(\mathbf{q}) = d_0(\mathbf{q}) \pm \sqrt{d_2^2 + d_3^2}$ and the band crossing points require $d_2(\mathbf{q}) = 0$ and $d_3(\mathbf{q}) = 0$. For a nodal ring centered at M , first, $d_2(\mathbf{q}) = \alpha_1 q_1 + \alpha_2 q_2 + \alpha_3 q_3 = 0$ can determine a plane passing the central point of the circle and the $d_3(\mathbf{q}) = b + \beta_1 q_1^2 + \beta_2 q_2^2 + \beta_3 q_3^2 = 0$ is an equation of an ellipsoidal surface centered at M . The crossings between the plane and the ellipsoidal surface form a closed loop, which is the nodal ring centered at M . However, this nodal can tilt to arbitrary direction. For the $\text{Pm}\bar{3}m$ space group, the mirror symmetry plays an important role for the nodal line (ring). Here, we can choose the mirror symmetry as $\hat{M}_z = \sigma_z$ to give an additional constraint to Hamiltonian,

$$H(\hat{M}_z \mathbf{q}) = \hat{M}_z H(\mathbf{q}) \hat{M}_z^{-1}$$

Using σ_z to replace \hat{M}_z , we can obtain the following expressions constrained by mirror symmetry

$$\begin{aligned} d_{1,2}(q_x, q_y, q_z) &= -d_{1,2}(q_x, q_y, 1 - q_z), \\ d_{0,3}(q_x, q_y, q_z) &= -d_{0,3}(q_x, q_y, 1 - q_z). \end{aligned}$$

Therefore, the M -centered nodal ring can be confined to the plane of $q_z = 0.5$. The same for the nodal lines along high symmetry line, other mirrors can also constrain them along one line as illustrated in Ref[17, 18].

Here, in system with PT symmetry, we want to point out that breaking the mirror symmetries should unlock the nodal lines from the mirror planes, but that the nodal lines (rings) will still exist, as they are protected by P and T . Ref.[17] has investigated the evolution of nodal lines under the inversion symmetry broken. **In our revised manuscript, we have added the detailed interpretation for this part.**

Comment #1.8: *The authors calculate the Chern number over a closed surface to identify a Weyl point. I wonder how the Chern number is numerically evaluated given that the wave functions usually suffer a global phase uncertainty.*

Authors' Reply: Thanks for this comment. In electronic system, one defined Chern number as a global phase uncertainty and calculated the Chern number of period plane in 3D BZ to reflect this global feature. Here, in phonon system, we usually defined the band Chern number (also called topological charge of WPs) to describe the locally sink or source feature of them as used in Refs. [1, 2, 14, 19, 20]. The computational detail is

that we defined a closed surface enclosing a Weyl point and integrated the Berry curvature over this surface as follows:

$$C_{\pm} = \frac{1}{2\pi} \oint \boldsymbol{\Omega}_{\pm} \cdot d\mathbf{S}$$

where $\boldsymbol{\Omega}_{\pm} = i\nabla_k \times \langle \psi_{\pm} | \nabla_k | \psi_{\pm} \rangle$ are the Berry curvatures and ψ_{\pm} are the wave functions of the two Blöch bands. This will give us C of $0, \pm 1$ or ± 2 .

Comment #1.9: In Fig. 2f, the authors plot the Wannier center evolution. In a torus, one can calculate the Wannier centers along one direction with respect to another momentum. Can the authors briefly explain how this is performed in a closed surface like a sphere? What is theta?

Authors' Reply: The method we use to calculate the evolution of Wannier center can be understood more vividly through the Figure 1.

FIG. 1. The Wilson-loop method for cylinder and sphere. Left: the discrete orbitals of a cylindrical surface. Center: the discrete orbitals of a spherical surface. Right: the evolution of phase ϕ represents winding number corresponding to Chern number.

When we consider the Wilson-link along q_x direction.

$$F_{i,i+1}^{mn}(k_y) = \langle m(k_{x,i}, k_y) | n(k_{x,i+1}, k_y) \rangle$$

For each q_y , we can define a product of $F_{i,i+1}$ as

$$D(k_y) = F_{0,1} F_{1,2} F_{2,3} \cdots F_{N_x-2, N_x-1} F_{N_x-1, 0}.$$

$H_w(k_y) = -i \log D(k_y)$ is the Wannier Hamiltonian[21]. The Wannier center evolution can be obtained from the phase evolution $\phi(k_y)$ of $D(k_y)$. The winding number of $\phi(k_y)$ of a closed surface can give the Chern number of band. Although this method can't give the value of Chern number (one need to count the winding number), this method can be easily applied to the cylinder, spherical surfaces or other closed surfaces. It can be used to analyze the topological charge or Chern number of nodal points or nodal line (rings). To be more detailed, for the case in our main text, we can construct a sphere centered at the WP, then get many discrete orbitals by the polar angle θ . For each orbital, we can construct the Wannier Hamiltonian H_w and get the phase ϕ . Finally, we can also count the winding number to give the Chern number.

Comment #1.10: *Other minor problems: “exists” in “there exists two kinds of high ...” should be “exist”. “The” in “For instance, The single type-I Weyl ...” should be “the”.*

Authors’ Reply: Thanks for your carefully reading and kind suggestions. We have corrected those two words and we also corrected other typos in in the manuscript and revised them one by one. All the revised sentences are marked in red.

Detailed responses to Referee #2:

Comment #2.1: *The authors did high-throughput searching topological states on phonon systems. Although the data screening took a lot of time, I think this paper is more suitable for Nature scientific data instead of Nature communication due to its lack of innovation. The new predicted materials are also not quite interesting to this community. So I don't recommend it to be published in Nature communication.*

Authors' Reply: We thank the referee very much for his/her nice summaries, suggestions and comments of our current manuscript.

First, concerning what the referee mentioned as “the data screening took a lot of time”, we would like to emphasize as follows,

i) To date, the phonon calculations of materials are still very expensive because large supercells need be used to get reliable force constants for phonon analysis. For this project, we have spent two whole years for phonon calculations by *VASP* and our undergoing calculations for the rest structures may last for at least two more years.

ii) In addition to the topological analysis, our phonon database contains tens of thousands of data for both structures and phonons, which can provide valuable data for both theory and experiments. The apparent advantages do not only save people much time of repeating the calculations of phonons, but also bring us much new knowledge on topological phononic materials, as outlined below.

iii) To date, there are not many topological phonon materials and, to our best knowledges, there are 20 publications since 2018. Recently, in 2020, there were already 10 publications (3 PRLs, 1 PRX and 6 PRBs) to predict the topological phonon in solid materials. Currently, there are many experimental groups (including our own collaborators) are trying to detect the topological phonons and related topological nontrivial surface states, in which the ideal topological phononic candidates are urgently needed. At current stage, the new predicted materials of topological phonons will attract a lot of people's attention and interests.

Our high-throughput calculations and database focus on the topological analysis, and it supplies a large amount of ideal topological phononic materials and the automatically topological analysis workflow (*HT-PHONON*) to this community. Of course, the theoretical predictions and experimental verifications will be the foundation for upcoming applications.

Second, concerning what the referee mentioned as “The new predicted materials are also not quite interesting to this community”, we would like to emphasize that the topological phonon is opening its door. These materials would be highly interesting for future applications. For the sake of clarity, I would like to mention the followings,

i) THz phonon of atomic lattice vibrations plays a key role in many important transport processes, such as heat conduction, heat-electricity energy conversion as well as electron-phonon coupling effects, and so on. The role of topological phonons on

these interesting transportation properties is still in the early stage of research.

ii) Topological phonons have potential applications in phonon wave-guides, thermoelectric, thermal isolations and other phononic devices. The occurrence of the topological phonons can be protected by various symmetries and their related topological non-trivial surface (edge) states are highly robust, against defects, impurities, and scatterings. All these features will play an important role in its further applications.

iii) our current work is advancing our understanding of topological phonons from the following four aspects,

- a) Topological phonons extensively exist in phonon spectra of many known materials, which can be classified into two basic types of Weyl and nodal-line (ring) topological phonons. Weyl phonons can be classified into single and high degenerate Weyl phonons. The former is always associated with the inversion symmetry broken, whereas the latter still requires the screw axis. Nodal-line (ring) phonons can be grouped into four subgroups of individual nodal-line (ring), nodal-link, nodal-chain and nodal-net phonons, upon the manipulation of the PT symmetry and the nonsymmorphic symmetry (e.g. screw axis or slide mirror or both). We have elucidated the physical mechanism for the occurrence of Weyl and nodal-line (ring) topological phonons. Furthermore, we have emphasized their mutual relationship among various types of topological phonons in new Fig. 6.
- b) Among phononic system, for the first time we have predicted the interesting hourglass nodal net topological phonons in TeO_3 (new Fig. 5). This kind of topological hourglass nodal net phonons is very special, which is highly robust due to the nonsymmorphic symmetry protection. The hourglass nodal net phonons can induce the topologically protected drumhead-like surface states. Due to the diversity of nonsymmorphic symmetry, more novel topological phonons can be indeed expected.
- c) Among phononic system, for the first time we have still found the clean, single and type-I Weyl phonons formed by the exact touching in the phononic acoustic-optical gap in half-Hesuler LiCaAs (Fig. 2). Physically, this existence of Weyl point associated with the zero acoustic-optical gap may significantly enhance the phononic scattering, as compared with the non-zero acoustic-optical gap phonons. This fact would have potential effect in reducing crystalline thermal conductivity. To our best knowledge, this Weyl phonon exhibits two unique features. It is the only reported type-I Weyl phonon, which is exactly formed by the touching between the highest acoustic branch and lowest optical branch. Meanwhile, it is also the only clean type-I Weyl phonon in the whole first Brillouin zone (BZ), no overlapping with any other phonon branches.
- d) Our results still reveal that the extensive coexistence of different types of topo-

logical phonons in materials, such as, the coexisted three-fold and four-fold degenerate Weyl phonons in unconventional superconductor of BeAu and the coexisted nodal-line and nodal-ring topological phonons in ScZn. The nonsym-morphic symmetry plays an important role for the formation of the high degenerate Weyl phonons and nodal-net topological phonons.

In summary, we hope the referee accept our work to be published in *Nature Communications*, because this manuscript does not only provide a comprehensive and systematical data for thousands of TPs, but also reveals new insights of physics and new types of topological phononic materials as discussed above.

Comment #2.2: *By the way, there is another question about the workflow shown in Fig1. How do you calculate the Chern number when you have nodal line? I mean the step between "P-T inversion symmetry?" to "To compute Chern number".*

Authors' Reply: Thanks for this comment. To be honest, this problem also puzzles us for a long time to classify the nodal rings and WPs in noncentrosymmetric materials. If we want to calculate the Chern number, we need to construct a closed surface to integrate the Berry curvatures ($\Omega_{\pm} = i\nabla_k \times \langle \psi_{\pm} | \nabla_k | \psi_{\pm} \rangle$) on it. However, this method needs separate wave functions at the whole closed surface. To avoid the degenerate points, we have two routes to solve this problem.

In the first, we construct a closed cubic centered at a crossing. By using the Stokes' theorem, $\oint_C dk A_n = \oint_S ds \nabla_k \times A_n = \oint_S ds \Omega$, we just need to calculate the integration of Berry connection along the boundaries of six cubic surfaces where we can avoid the degeneracy problem. For isolated nontrivial crossings, we will get an integer Chern number. For crossing points on the nodal line (rings), we will get zero Chern number.

In the second, we used the conjugate gradient optimization algorithm to identify continuous lines and isolated points by judging whether there are nearby points within a small area. For isolated points, we calculate the Chern number by integrating the Berry curvatures on the closed surface of a small enough sphere. For nodal line (rings), each crossing point on the continuous wire has the nonzero Berry phase and they can be classified into the topological nodal lines and topological nodal rings depending on their shapes. Those two classification methods we used agree well with each other and can get mutually consistent results.

We have emphasized the above two methods, which were exactly adopted in our current HT-phonon tool, in our revised manuscript.

Finally, we would like to thank the referee again very much for his/her careful reading, nice suggestions and comments. Here, by taking this chance again, we would like to emphasize that the phonon calculations consume a lot of expensive computing resources and precious time. It is the first huge work focusing on the phonons and topological phonons of solid materials with periodic atomic lattices.

We hope that this work can serve as a platform for a broad research field related to phonon and its underlying physics including topological phonons and transport for both experimental and theoretical studies. In addition, we believe that this work would attract much more attentions and interests even in the fields of topological superconductor[3, 22], high performance thermoelectricity[23, 24], and novel phononic devices[25, 26].

Detailed responses to Referee #3:

Comment #3.1: *The authors have presented a systematic investigation and algorithm for the search of topological phononic states in solids with over 10^4 materials with 5014 topological phononic materials identified. Some important classes and examples are elaborated. This work is impressive. However, I have several comments:*

Authors' Reply: We thank the referee very much for his/her nice summaries, suggestions and comments of our current manuscript. As your summaries, “*over 10^4 materials with 5014 topological phononic materials identified*”, we indeed took a long time for this project and overcame many difficulties of both theory and technology to realize the specialized topological phonons with a large amount of data. Importantly, we are still performing this project, which will last for the next two years for the remaining structures in our hands.

We hope this work can supply a platform to provide objective data in a broad research field related to phonon transport for both experiment and theory studies. Thanks again for your recognition of our work.

Comment #3.2: *I basically think the manuscript is too technical which narrows the scope of audience. The authors could write it in a more transparent, physical and materials way, trying to state the findings in plain languages and avoid using jargons as much as possible.*

Authors' Reply: We thank the referee very much for this valuable suggestion. We agree that the last version of our manuscript is too technical and some implementation details are not transparent. Hence, we have spent a long time to polish our manuscript and almost rewrote the full main text and added many supporting evidences for potential applications in plain languages as your suggestions. Hopefully, this revised current manuscript are more approachable and easy to read, which will attract interests of more audiences.

Comment #3.3: *The examples are discussed too briefly (Nature Commun. allows much more space than the length of the current manuscript), making the main body of the manuscript less attractive.*

Authors' Reply: We thank the referee very much for this reminder. As your suggestion, for those four representations of topological phonons and we have also added many contents to discuss the results of those examples both in models and potential applications. All the changed contents are marked as red in our manuscript.

Comment #3.4: *Most importantly, for the examples analyzed by the authors, the generic underlying physics is unclear especially those which are unique to phonons.*

Authors' Reply: Thanks for your nice suggestions. In our revised manuscript, we have added detailed discussions and underlying physics for each examples to illustrate the occurrence of single Weyl phonons in LiCaAs, of high degenerate Weyl phonons in BeAu, of coexisted nodal-line and nodal-ring topological phonons in ScZn and of the new type of hourglass nodal net topological phonons in TeO₃ from the analysis of both symmetry and $k\cdot p$ model. In addition, we have elucidated their correlations and relationship of various different topological phonons in new Fig. 6.

Of course, those four materials are just tips of the iceberg and the supplementary materials already supplied a large amount of prototypes of topological phonons for studies to dig more underlying physics, such as the complex combinations of novel excitations, transformations of topological phonons under symmetry broken, and relations to transport process.

Comment #3.5: *I strongly suggest the following revisions:*

Authors' Reply: Thank you much for your suggestions to improve our manuscript. Due to the emerging developing of topological phonons and the lack of efficient control methods, it's much weaker of its applications. But there are still some researches to supply many wonderful and practical concepts for high performance thermoelectric materials, topological superconductivity, and novel phonon devices in thermal transportations and photon-phonon interactions. There are many detections methods for both bulk phonons and surface phonons and those probing techniques have developed rapidly recently. We have summarized these methods in details. And for mechanical metamaterials we have also mentioned them in our main text.

Comment #3.6: *Possible physical properties of topological phonons, e.g., optical or transport properties, must be discussed in as much details as the author can provide.*

Authors' Reply: Thanks for this kind suggestions. Topological phonons can be utilized in many aspects. We have added a paragraph in our main text to give the potential applications of topological phonons. First, it's important for fundamental physics. Analogy to electrons, many quantum states can be expected in topological phonon such as quantum valley Hall effect (QVHE) and quantum (anomalous/spin) Hall states (Q(A/S)H). Second, protected one-way edges states, which are immune to scattering from defects, can be used to realize some exciting applications such as thermal diodes and waveguides, thermal rectification and the novel bulk phonon can be used to realize the phonon filter, phononic antennas. Third, under some conditions, topological bulk phonon can provide more phonon scattering channels in the three phonon-phonon scattering processes and suppress the lattice thermal conductivity, which can be used to enhance the thermoelectric response. Fourth, combining the nontrivial surface states of both phonons and electrons, the electron-phonon (el-ph) coupling may be enhanced and the anomaly superconductivity can be expected at surfaces and interfaces. For example,

as shown in Fig.2, the work we are working on shows that the enhance of el-ph coupling can be expected from 10~15THz of nontrivial surface phonon in topological Dirac nodal line metal Be. Finally, in the future, we may take great advantage of controlling the phonons to manipulate heat and sound as electrons and photons and even make a significant technological revolution in our society. In the revised manuscript, we have added those discussions in details.

FIG. 2. DFT-derived electronic band structure (up panel) and phonon dispersion (down panel) of the Be (0001) surface. The up panel: Surface electronic structures along the high symmetry lines as compared with available ARPES experimental data. The down panel: Surface phonon dispersion along the high symmetry lines compared with available experimentally observed surface phonon dispersions[27]. The left of each panel: Surface density of states (DOSs) of electrons and phonons, respectively.

Comment #3.7: *Possible detections of topological phonons must be discussed.*

Authors' Reply: Thanks for this kind suggestions. Due to the disappearance of Pauli exclusion principle, phonon spectrum can be physically probed at the whole frequency range. There are many experimental technologies for phonon detestations, such as inelastic x-ray scattering (IXS) and electron energy loss spectroscopy (EELS) etc. As shown in Fig.2, the surface phonon of Be metal have been probed by EELS in 1990s[27, 28]. And we have discussed those methods and scopes of application in our main text. Based on those methods, we wish more topological phonons both of bulk and surface can be detected and this will greatly promote the development of topological phonon applications.

Comment #3.8: *Topological phonon properties are also discussed in mechanical metamaterials, e.g., Nature volume 555, pages342–345(2018); Phys. Rev. B 97, 180101(R), These works should be mentioned.*

Authors' Reply: Thank you very much for this important suggestion. We are sorry to missing the important part of mechanical metamaterials and topological phonon in this field has many new developments on various verifications of topological phases and phonon applications, such as acoustic/heat cloaking and thermal metamaterials of heat management. We have discussed them in details in our main text by citing more references, also including the papers the referee mentioned above.

Comment #3.9: *In addition, Refs. 33 and 46 are redundant.*

Authors' Reply: Thanks for your carefully reading and this kind suggestion. We have deleted the redundant reference.

References:

1. Zhang, T., et al., *Double-Weyl Phonons in Transition-Metal Monosilicides*. Phys Rev Lett, 2018. **120**(1): p. 016401.
2. Li, J., et al., *Coexistent three-component and two-component Weyl phonons in TiS, ZrSe, and HfTe*. Physical Review B, 2018. **97**(5).
3. Li, R., et al., *Underlying Topological Dirac Nodal Line Mechanism of the Anomalously Large Electron-Phonon Coupling Strength on a Be (0001) Surface*. Phys Rev Lett, 2019. **123**(13): p. 136802.
4. Soluyanov, A.A., et al., *Type-II Weyl semimetals*. Nature, 2015. **527**(7579): p. 495-8.
5. Vergniory, M.G., et al., *A complete catalogue of high-quality topological materials*. Nature, 2019. **566**(7745): p. 480-485.
6. Tang, F., et al., *Comprehensive search for topological materials using symmetry indicators*. Nature, 2019. **566**(7745): p. 486-489.
7. Zhang, T., et al., *Catalogue of topological electronic materials*. Nature, 2019. **566**(7745): p. 475-479.
8. Po, H.C., A. Vishwanath, and H. Watanabe, *Symmetry-based indicators of band topology in the 230 space groups*. Nat Commun, 2017. **8**(1): p. 50.
9. Khalaf, E., et al., *Symmetry Indicators and Anomalous Surface States of Topological Crystalline Insulators*. Physical Review X, 2018. **8**(3).
10. Kruthoff, J., et al., *Topological Classification of Crystalline Insulators through Band Structure Combinatorics*. Physical Review X, 2017. **7**(4).
11. Bradlyn, B., et al., *Topological quantum chemistry*. Nature, 2017. **547**(7663): p. 298-305.
12. Mañes, J.L., *Fragile phonon topology on the honeycomb lattice with time-reversal symmetry*. Physical Review B, 2020. **102**(2).
13. Liu, Q.-B., et al., *Symmetry-enforced Weyl phonons*. npj Computational Materials, 2020. **6**(1).
14. Wang, R., et al., *Symmetry-Protected Topological Triangular Weyl Complex*. Phys Rev Lett, 2020. **124**(10): p. 105303.
15. Zhang, T.T., et al., *Phononic Helical Nodal Lines with PT Protection in MoB₂*. Phys Rev Lett, 2019. **123**(24): p. 245302.
16. Burkov, A.A., M.D. Hook, and L. Balents, *Topological nodal semimetals*. Physical Review B, 2011. **84**(23).
17. Li, J., et al., *Phononic Weyl nodal straight lines in MgB₂*. Physical Review B, 2020. **101**(2).
18. Liu, Q.-B., et al., *Categories of Phononic Topological Weyl Open Nodal Lines and a Potential Material Candidate: Rb₂Sn₂O₃*. The Journal of Physical Chemistry Letters, 2019. **10**(14): p. 4045-4050.
19. Xia, B.W., et al., *Symmetry-Protected Ideal Type-II Weyl Phonons in CdTe*. Phys Rev Lett, 2019. **123**(6): p. 065501.
20. Miao, H., et al., *Observation of Double Weyl Phonons in Parity-Breaking FeSi*. Phys Rev Lett, 2018. **121**(3): p. 035302.
21. Benalcazar, W.A., B.A. Bernevig, and T.L. Hughes, *Quantized electric multipole insulators*.

- Science, 2017. **357**(6346): p. 61.
22. Jin, K.-H., et al., *Topological superconducting phase in high- T_c superconductor MgB_2 with Dirac-nodal-line fermions*. npj Computational Materials, 2019. **5**(1).
 23. Snyder, G.J. and E.S. Toberer, *Complex thermoelectric materials*. Nature Materials, 2008. **7**(2): p. 105-114.
 24. Singh, S., et al., *Topological phonons and thermoelectricity in triple-point metals*. Physical Review Materials, 2018. **2**(11).
 25. Wang, H., et al., *Experimental study of thermal rectification in suspended monolayer graphene*. Nature Communications, 2017. **8**(1): p. 15843.
 26. Liang, B., et al., *An acoustic rectifier*. Nature Materials, 2010. **9**(12): p. 989-992.
 27. Hannon, J.B. and E.W. Plummer, *Shear horizontal vibrations at the (0001) surface of beryllium*. Journal of Electron Spectroscopy and Related Phenomena, 1993. **64-65**: p. 683-690.
 28. Hofmann, P. and E.W. Plummer, *Lattice vibrations at the $Be(10\bar{1}0)$ surface*. Surface Science, 1997. **377-379**: p. 330-334.

REVIEWER COMMENTS

Reviewer #1 (Remarks to the Author):

The authors have addressed almost all my comments and concerns in the reply and the substantially revised manuscript, which I really appreciate. In the revised paper, the authors have also predicted the hourglass nodal net topological phonons in TeO₃, besides providing an almost complete catalog of topological nodal points and nodal lines (rings) for phononic materials. The work is now impressive and would pave the way for further study of topological phonons in materials. Yet, before I can recommend the work for publication in Nature Communications, I still have a technical question that the authors have not addressed in their reply.

For my previous comment 7, I mean when you calculate the Chern number through the formula $C = \oint \Omega \cdot dS$ with $\Omega = i \nabla_{\mathbf{k}} \times \langle \psi | \nabla_{\mathbf{k}} | \psi \rangle$, if the Berry curvature is calculated through numerical differentiation of wave functions, it is usually hard to be achieved because the wave functions obtained numerically usually have a phase uncertainty. I wonder how the authors solve this difficulty. From the reply to Comment 2 raised by Referee 2, I see that the integration of Berry connection along the boundaries of six cubic surfaces are actually calculated instead. But the Stokes' theorem can be used when the Berry connection is a continuously differentiable vector field, which requires an appropriate gauge of wave functions. Can the authors explain how the calculation is performed?

Another minor question: In the first paragraph of page 7, when g_1 is applied to $|u_q\rangle$, why the image vector is still $|u_q\rangle$ rather than $|u_{\{qx, qy, -qz\}}\rangle$?

Reviewer #3 (Remarks to the Author):

I find that the revised manuscript has been considerably improved, both in its presentation and the contents. Although the presentation has not reached to the ideal level for Nature Communications. The contents are much clearer and the results are much more attractive and significant. In particular, the several prototype cases elaborated in the main text are very interesting and important. For this reason, I recommend the publication of this paper and I firmly believe that the value of the paper is beyond its appearance.

Response to Referees' report
(manuscript NCOMMS-20-23399)

Detailed responses to Referee #1:

Comment #1.1: *The authors have addressed almost all my comments and concerns in the reply and the substantially revised manuscript, which I really appreciate. In the revised paper, the authors have also predicted the hourglass nodal net topological phonons in TeO3, besides providing an almost complete catalog of topological nodal points and nodal lines (rings) for phononic materials. The work is now impressive and would pave the way for further study of topological phonons in materials. Yet, before I can recommend the work for publication in Nature Communications, I still have a technical question that the authors have not addressed in their reply.*

Authors' Reply: We thank the referee very much for his/her nice summaries, suggestions and comments of our manuscript. We have carefully revised main contents under your last suggestions and comments to improve the quality of our manuscript and we are very happy to hear your positive comments for our research.

Comment #1.2: *For my previous comment 7, I mean when you calculate the Chern number through the formula $C = \oint \Omega \cdot dS$ with $\Omega = i \nabla_k \times \langle \psi | \nabla_k | \psi \rangle$, if the Berry curvature is calculated through numerical differentiation of wave functions, it is usually hard to be achieved because the wave functions obtained numerically usually have a phase uncertainty. I wonder how the authors solve this difficulty. From the reply to Comment 2 raised by Referee 2, I see that the integration of Berry connection along the boundaries of six cubic surfaces are actually calculated instead. But the Stokes' theorem can be used when the Berry connection is a continuously differentiable vector field, which requires an appropriate gauge of wave functions. Can the authors explain how the calculation is performed?*

Authors' Reply: Thanks for raising this question. We have used the numerical method described by Fukui *et al.* in J. Phys. Soc. Jpn. 74, 1674 (2005) to calculate the Chern number. In this paper, a gauge independent field strength is defined as [see Eq. (8) in JPSJ 74, 1674(2005)]

$$\tilde{F}_{12}(k_l) \equiv \ln U_1(k_l) U_2(k_l + \hat{1}) U_1(k_l + \hat{2})^{-1} U_2(k_l)^{-1}$$

Here U_μ is a U(1) link variable

$$U_\mu \equiv \langle n(k_l) | n(k_l + \hat{\mu}) \rangle / |\langle n(k_l) | n(k_l + \hat{\mu}) \rangle|$$

The summation of field strength on discrete k meshes gives rise to the Chern number [see Eq. (9) in JPSJ 74, 1674(2005)]:

$$\tilde{c}_n = \frac{1}{2\pi i} \sum_l \tilde{F}_{12}(k_l) = \sum_l n_{12}(k_l)$$

Therefore, the Chern number is also gauge independent and any choice of gauge gives

an identical Chern number[1]. Similarly, the Berry phase is also gauge independent [2, 3]

$$\gamma_n = \oint A_n dl = -i \ln[\langle \phi_n^0 | \phi_n^1 \rangle \langle \phi_n^1 | \phi_n^2 \rangle \cdots \langle \phi_n^{N-1} | \phi_n^0 \rangle]$$

Which also means any choice of gauge gives an identical Berry phase. The essence is that as long as the topological invariances are defined in a gauge independent manner, any choice of gauge can be adopted for numerical calculations. It will give gauge independent results. Therefore, the phase uncertainty can be solved by above methods.

Comment #1.3: *Another minor question: In the first paragraph of page 7, when \$gI\$ is applied to \$|u_q\rangle\$, why the image vector is still \$|u_q\rangle\$ rather than \$|u_{\{qx,qy,-qz\}}\rangle\$?*

Authors' Reply: Thanks for your careful reading. If operator \$g\$ is a symmetry of the crystal, the Hamiltonian should satisfy the equation:

$$H(\hat{g}\mathbf{q}) = \hat{g}H(\mathbf{q})\hat{g}^{-1}$$

Assuming the eigen-equation

$$H(\mathbf{q})u(\mathbf{q}) = \lambda u(\mathbf{q})$$

Then

$$\begin{aligned} \hat{g}H(\mathbf{q})\hat{g}^{-1}\hat{g}u(\mathbf{q}) &= \lambda\hat{g}u(\mathbf{q}) \\ H(\hat{g}\mathbf{q})\hat{g}u(\mathbf{q}) &= \lambda\hat{g}u(\mathbf{q}) \end{aligned}$$

For \$\hat{g}\mathbf{q}\$ on the invariant line or plane that satisfies \$\hat{g}\hat{g}=\mathbf{q}\$, we have

$$H(\hat{g}\mathbf{q})\hat{g}u(\mathbf{q}) = \lambda\hat{g}u(\mathbf{q})$$

We see that \$\hat{g}u(\mathbf{q})\$ is also the eigenstate of \$H(\hat{g}\mathbf{q})\$ with eigenvalue \$\lambda\$. Therefore, \$\hat{g}u(\mathbf{q})\$ differs from \$u(\hat{g}\mathbf{q})\$ by a phase factor. Finally, when \$g\$ is applied to \$u(\mathbf{q})\$, then image vector is still \$u(\mathbf{q})\$. The phase factors can be regarded as the eigenvalues of the symmetry operation [4] such as our familiar mirror eigenvalues[5] or glide mirror eigenvalues[6, 7].

In our main text, we revised the sentence for this equation to avoid misunderstandings.

“Under the glide reflection operation \$g_1\$, for invariant lines or planes satisfying \$g_1^2=\mathbf{q}\$, the Bloch states can be eigenstates \$g_1|u(\mathbf{q})^\pm\rangle = \pm\lambda e^{iq\cdot t} |u(\mathbf{q})^\pm\rangle\$.”

Detailed responses to Referee #3:

Comment #3.1: *I find that the revised manuscript has been considerably improved, both in its presentation and the contents. Although the presentation has not reached to the ideal level for Nature Communications. The contents are much clearer and the results are much more attractive and significant. In particular, the several prototype cases elaborated in the main text are very interesting and important. For this reason, I recommend the publication of this paper and I firmly believe that the value of the paper is beyond its appearance.*

Authors' Reply: We thank the referee very much for his/her nice summaries and comments of our manuscript. We hope this manuscript can pave the way for further study of topological phonons in materials and can be accepted by *Nature Communications*.

References:

1. Fukui, T., Y. Hatsugai, and H. Suzuki, *Chern Numbers in Discretized Brillouin Zone: Efficient Method of Computing (Spin) Hall Conductances*. Journal of the Physical Society of Japan, 2005. **74**(6): p. 1674-1677.
2. Victor, B.M., *Quantal phase factors accompanying adiabatic changes*. Proc. R. Soc. Lond. A, 1984. **392**: p. 45-57.
3. Vanderbilt, D., *Berry Phases in Electronic Structure Theory_ Electric Polarization, Orbital Magnetization and Topological Insulators*. Cambridge University Press, 2018.
4. Parameswaran, S.A., et al., *Topological order and absence of band insulators at integer filling in non-symmorphic crystals*. Nature Physics, 2013. **9**(5): p. 299-303.
5. Bzdusek, T., et al., *Nodal-chain metals*. Nature, 2016. **538**(7623): p. 75-78.
6. Wang, Z., et al., *Hourglass fermions*. Nature, 2016. **532**(7598): p. 189-94.
7. Young, S.M. and C.L. Kane, *Dirac Semimetals in Two Dimensions*. Phys Rev Lett, 2015. **115**(12): p. 126803.

REVIEWERS' COMMENTS

Reviewer #1 (Remarks to the Author):

The authors have addressed all my questions. I can now recommend the manuscript for publication in Nature Communications.